



# Model predictions of long-lived storage of organic carbon in river deposits

Mark A. Torres[1], Ajay B. Limaye[2], Vamsi Ganti[3], Michael P. Lamb[1], A. Joshua West[4], and Woodward W. Fischer[1]

[1]Division of Geological & Planetary Sciences, California Institute of Technology
[2]Department of Earth Sciences, University of Minnesota, 2 SE 3rd Ave., Minneapolis, MN 55414
[3]Department of Earth Science & Engineering, Imperial College
[4]Department of Earth Sciences, University of Southern California

*Correspondence to:* Mark A. Torres (mtorres@caltech.edu)

**Abstract.**

The mass of carbon stored as organic matter in terrestrial systems is sufficiently large to play an important role in the global biogeochemical cycling of $CO_2$ and $O_2$. Field measurements of radiocarbon-depleted particulate organic carbon (POC) in rivers suggest that terrestrial organic mat-

ter persists in surface environments over millennial (or greater) timescales, but the exact mechanisms behind these long storage times remain poorly understood. To address this knowledge gap, we developed a numerical model for the radiocarbon content of riverine POC that accounts for both the duration of sediment storage in river deposits as well as the effects of POC cycling. We specifically target rivers because sediment transport defines the maximum amount of time organic matter

can persist in the terrestrial realm and river catchment areas are large relative to the spatial scale of variability in biogeochemical processes.

   Our results show that rivers preferentially erode young deposits, which, at steady-state, requires that the oldest river deposits are stored for longer than expected for a well-mixed sedimentary reservoir. This geometric relationship can be described by an exponentially-tempered power-law distri-

bution of sediment storage durations, which allows for significant aging of biospheric POC. While OC cycling partially ameliorates the effects of sediment storage, the consistency between our model predictions and a compilation of field data highlights the important role of storage in setting the radiocarbon content of riverine POC. The results of this study imply that the controls on the terrestrial OC cycle are not limited to the factors that affect rates of primary productivity and respiration, but

also include the dynamics of terrestrial sedimentary systems.





## 1 Introduction

Terrestrial organic matter present in the biosphere, soils, and other shallow sedimentary deposits represents an enormous reservoir of carbon (4 to $5 \times 10^{12}$ tonnes C; Fischlin et al., 2007) whose dynamics influence atmospheric $O_2$ and $CO_2$ concentrations over annual (Keeling, 1960; Keeling

and Shertz, 1992; Stallard, 1998) to geologic timescales (Bird et al., 1994; France-Lanord and Derry, 1997). Understanding the links between terrestrial organic carbon (OC) cycling and atmospheric $O_2$ and $CO_2$ concentrations requires knowledge of the timescales over which OC persists in the environment before being oxidized (Berner, 1989) as well as the underlying processes that set these timescales. Identifying these processes and timescales is challenging because existing measurements

of OC lifetimes show scale-dependence (i.e., the measured rate of organic carbon oxidation decreases with increasing measurement timescale; Middelburg 1989; Katsev and Crowe 2015). Consequently, measurements of terrestrial OC cycling need to made at large spatial and temporal scales to be quantitatively linked to global biogeochemical cycles.

Large rivers integrate the dynamics of OC cycling at spatial and temporal scales large enough to

35 relate to global biogeochemical cycles. Rivers transport particulate OC (POC) eroded from across their catchment areas to the ocean, where POC is either oxidized or buried in marine sediments (Blair and Aller, 2012). Thus, rivers set the amount of time POC can persist within the terrestrial realm and integrate over areas that are large compared to the spatial scales of variability in biogeochemical processes. Annually, rivers transport large masses of POC to the ocean (1 to $2 \times 10^8$ tonnes C yr$^{-1}$;

Galy et al., 2015), making fluvial transport a relevant carbon cycle flux over a range of timescales.

At the scale of river catchments, radiocarbon ($^{14}$C) provides a natural tracer of the lifetime of OC in surface environments. Previous studies have measured large variations in the $^{14}$C/$^{12}$C of riverine POC (Masiello and Druffel, 2001; Martin et al., 2013; Clark et al., 2013; Tao et al., 2015), including significant variations in $^{14}$C/$^{12}$C with depth in river channels (Galy et al., 2008; Bouchez

et al., 2014). Direct interpretation of measured $^{14}$C/$^{12}$C in terms of ages would imply that most riverine POC is old (thousands to tens of thousands of years). However, much of the variation in $^{14}$C/$^{12}$C has been interpreted in the context of variable admixtures of two carbon components: radiocarbon-bearing organic compounds synthesized within the river catchment ("biospheric" POC) and radiocarbon-dead organics derived from the erosion of ancient sedimentary rocks ("petrogenic"

POC; Masiello and Druffel 2001; Galy et al. 2008).

The erosion of petrogenic POC need not affect the atmospheric budgets of $CO_2$ and $O_2$ unless it is oxidized during transit (Bouchez et al., 2010), which releases $CO_2$ to and consumes $O_2$ from the atmosphere. Conversely, biospheric POC is a sink for $CO_2$ and source of $O_2$ over the course of its lifetime, which is thought to vary widely between different compound classes and environments

(Schmidt et al., 2011). While the lifetime of biospheric POC should be reflected in its $^{14}$C/$^{12}$C, determining this ratio from bulk radiocarbon measurements of riverine sediments may be complicated





by the mixing of biospheric POC ($^{14}$C/$^{12}$C $\leq$ atmospheric ratio) with petrogenic POC ($^{14}$C/$^{12}$C = 0).

Compound-specific radiocarbon analyses of terrestrial biomarkers, which can be interpreted as dominantly reflecting the biospheric component, suggest that biospheric POC has a substantially lower $^{14}$C/$^{12}$C than the atmosphere in some large river systems (Galy and Eglinton, 2011; Feng et al., 2013; Tao et al., 2015; Schefuß et al., 2016). These observations require that some biospheric POC is stored in terrestrial reservoirs over (at least) millennial timescales before being transported to ultimate depocenters in marine basins. It is reasonable to ask if these observations of aged biospheric POC in modern rivers result, at least in part, from transient storage in river sediment deposits (Galy and Eglinton, 2011; Feng et al., 2013; Tao et al., 2015; Schefuß et al., 2016).

To investigate how storage in river deposits influences the $^{14}$C content of riverine POC, we developed a numerical model that explicitly accounts for the effects of both sediment storage and OC cycling. The sediment storage component of our model is focused on how the stochastic nature of fluvial processes leads to the preferential recycling of young sediment deposits (Nakamura and Kikuchi, 1996; Bradley and Tucker, 2013), which, at steady-state, requires that the oldest floodplain deposits are stored for longer than expected for a well-mixed sedimentary reservoir (Bolin and Rodhe, 1973; Bradley and Tucker, 2013). This geometric relationship engenders a heavy-tailed (power-law) distribution of sediment storage durations that may allow for the significant aging of biospheric POC during storage in river deposits and impart a characteristic shape to the distribution of POC ages measured in rivers. However, the production and consumption of POC during floodplain storage can modify POC ages and may serve to ameliorate the effects of sediment storage on the POC age distribution. By including both fluvial and biogeochemical processes, our approach provides new insights into the interpretation of the terrestrial OC cycle and the radiocarbon content of riverine POC.

Starting from a generic theory for predicting the duration of sediment storage in river systems (Section 2.1), we developed a sediment storage model based on the dynamics of meandering rivers (Sections 2.2 and 2.3). We coupled this sediment storage model to a biogeochemical cycling model (Section 2.4) in order to build a full model for predicting the radiocarbon content of riverine POC under different sedimentary and biogeochemical scenarios. Predictions generated using this coupled model were compared with a compilation of field data from diverse global sites, demonstrating that the dynamics of sediment storage in shallow deposits have significant predictive skill in explaining–at least in part–the radiocarbon content of organic matter observed within many rivers (Section 3.3.1).





## 2  Model development

### 2.1  Generic theory for organic carbon and sediment storage

After being eroded from an upland source, fluvial sediments are routed through a transport network that includes multiple temporary storage reservoirs (e.g., channel and floodplain deposits; Malmon et al., 2003; Lauer and Parker, 2008a; Lauer and Willenbring, 2010; Pizzuto et al., 2016). These sediment reservoirs can also store POC, potentially leading to a decrease in its radiocarbon content due to radioactive decay. The magnitude of this effect will depend on: 1) the duration of sediment storage and 2) the rate at which the $^{14}$C content of POC changes with time. Developing simple models for these two factors forms the basis of our approach.

Sediment grains will spend some portion of their time in transient storage within sediment deposits and the remainder of their time in active transport associated with the river channel. We posit that the time spent in storage is much greater than the time spent in active transport (e.g., Sadler, 1981; Ganti et al., 2011) and, as a result, that the total transit time of sediments from source to sink can be approximated as the total time spent in storage. Since sediment grains likely enter and exit temporary storage reservoirs multiple times during transit, the total storage time can be separated into two components: 1) the number of times that grains enter and exit storage reservoirs and 2) and the duration of each storage event.

Fluvial processes are expected to cause natural variations in the duration of each storage event. This variability can be simulated by representing storage duration as a random variable with some probability distribution $p_S(t)$. Thus, for some number of storage events $n$, the transit time can be calculated as the sum of $n$ draws from the probability distribution of storage durations. Since the sediment load transported by rivers is composed of a collection of sediment grains with different storage histories, it will be characterized by some distribution of transit times that reflects both $p_S(t)$ as well as the total number of storage events. Mathematically, the transit time distribution ($p_{Tr}(t)$) is given by the convolution of the storage distribution by itself $n$ times

$$p_{Tr}(t) = p_S(t)^{*n} \qquad (1)$$

To predict the change in the radiocarbon content of POC that results from sediment storage, the functional relationship between the radiocarbon content of POC and time, $f(t)$, can be used to transform the transit time distribution into a distribution of radiocarbon contents ($p_F(^{14}C)$) where:

$$p_F(^{14}C) = p_{Tr}(f^{-1}(t)) \frac{dt}{d^{14}C} \qquad (2)$$

We note that $f(t)$ need not be the radioactive decay equation if POC is continuously produced and consumed during fluvial transit, which is considered in greater detail in Section 2.4.2.



## 2.2 Application of a meandering model to determine storage duration

Identifying appropriate mathematical expressions for the probability distribution of storage durations is critical to developing realistic models for several geochemical tracers in river sediments, including the radiocarbon content of POC, that have nonlinear changes in concentration with time. Most previous models of sediment storage in meandering river systems have assumed that storage durations are exponentially distributed (Malmon et al., 2003; Lauer and Parker, 2008a; Lauer and Willenbring, 2010). This assumption requires that deposits of all ages are equally likely to be eroded at any given time, which is inconsistent with some field data (Nakamura and Kikuchi, 1996; Lancaster and Casebeer, 2007) as well as the common observation that the position of the river channel is persistent in time (Bradley and Tucker, 2013). To develop a more realistic storage duration distribution, it is necessary to consider the physical processes that govern sediment exchange in natural river systems.

In lieu of following a static path, many rivers migrate laterally across their floodplains with time. Single-thread, migrating (i.e., meandering) channels are one of the most common channel types and show morphodynamic processes common to all alluvial rivers (Dunne and Aalto, 2013). Over time, lateral migration of single-thread rivers leads to the development of arcuate meander bends. Meander bends can grow until a cutoff event occurs where the river bypasses an existing portion of its reach in favor of a straighter path (e.g., Hooke, 1995). Together, lateral migration and cutoff allow rivers to traverse back and forth across their floodplains over time and continuously exchange sediments in active transport with those in passive storage. Thus, the spatial and temporal patterns of lateral migration and cutoff directly impact the duration of sediment storage in fluvial systems (Bradley and Tucker, 2013).

In order to capture the full range of relevant time and space scales, which are inaccessible in field observations of lateral migration (Black et al., 2010; Constantine et al., 2014), we used an existing numerical model of river meandering (Howard and Knutson, 1984; Limaye and Lamb, 2013) to derive a process-based probability distribution of storage durations. The meandering model assumes a fixed channel width and represents the channel position using a series of discrete nodes. At each node, local rates of relative channel migration depend upon the local and upstream-weighted channel curvature. Local rates of lateral migration are then computed from the relative channel migration rate, sinuosity, and a user-defined bank erodibility coefficient, which sets average lateral migration rate for each simulation. Neck cutoffs occur whenever the channel intersects itself; chute cutoffs are not modeled. We fixed the maximum channel lateral erosion rate at 0.05 channel widths per year, which is typical for actively migrating meandering rivers (e.g., Hickin and Nanson, 1975; Constantine et al., 2014), and ran each model simulation for a total of $10^5$ years. In the model topology, the channel is inset in a planar floodplain surface and migrates laterally with no bed elevation change. We neglect over bank deposition and the loss of sediment due to subsidence, but discuss these in Section 4.1. A schematic of the model is shown in Figure 1.





The numerical model runs yield a time-series of channel positions. For all timesteps, we define areas within the active channel as having a sediment age of zero. As time proceeds and the channel migrates, it abandons sediments along the inner bank (i.e., point bar deposits), which begin to age. When the active channel overlaps existing deposits, the time elapsed since the emplacement of those deposits is recorded as the storage duration. Since areas bounded by the channel are defined as having a sediment age of zero, the model does not allow for cumulative aging as a result of multiple deposition and transport events (i.e., once material is eroded, its age is reset to zero). As a consequence, the model does not measure the total amount of time sediments spend in storage (transit time), but instead tracks the amount of time spent in storage for a single deposition/erosion event (storage duration).

Using the time-series of channel positions, we calculated storage durations as the ages of river deposits that are eroded by lateral migration at each time step. The storage duration for each eroded deposit is weighted by its areal extent along the active channel, and observations from each timestep were combined to yield a full distribution of storage durations for the model. We note that excluding data from the beginning (e.g., the first 50%) of the model runs does not significantly affect the full distribution since short storage durations are more probable overall. We also examined the age distribution of deposits remaining on the floodplain at the end of the model run, which we termed the deposit age distribution.

Our approach of using a numerical model of river meandering is similar to that of Bradley and Tucker (2013), who developed a quasi-static storage duration distribution from a single meandering model simulation. The river meandering model used in this study (Howard and Knutson, 1984) differs from the Lancaster and Bras (2002) model used by Bradley and Tucker (2013). A salient difference between the models is that only the Lancaster and Bras (2002) model has been shown to develop compound bends prior to cutoff. However, similar bend geometries rapidly develop in the Howard and Knutson (1984) model due to meander cutoff. Also, compared to Bradley and Tucker (2013), our analysis uses a suite of model runs with boundary conditions that allow freer channel motion (e.g., the model uses a fixed upstream boundary condition, which limits drift of the mean channel axis; Limaye and Lamb, 2013). To systematically account for the sensitivity of the meandering model to the initial conditions, which can impact modeled river trajectories (e.g., Frascati and Lanzoni, 2009), we used five replicate simulations to determine storage duration distributions and fifty replicate simulations to determine deposit age distributions.

### 2.2.1 Scaling model results to natural systems

Our meandering model simulations should capture the appropriate shape of the storage duration distribution. However, absolute values should vary as a function of the lateral migration rate. Specifically, rivers that migrate quickly should have, on average, shorter storage durations relative to rivers that migrate slowly. In order to account for the effects of variable migration rates, we normalized all





storage durations by the time required for the channel to migrate laterally to the point of meander
bend cutoff ($T_{cut}$). By explicitly tracking bends from growth to cutoff (Schwenk et al., 2015), we
find that $T_{cut}$ is equal to 350 years in our meandering model simulations. Normalizing the meander-
ing model results by this $T_{cut}$ value yields a dimensionless storage duration distribution that can be
re-scaled to produce variable sediment storage times by varying $T_{cut}$.

To aid in model-data comparisons, it is useful to develop a prediction for how $T_{cut}$ may vary
between river systems. Based on a separate set of meandering model runs with variable maximum
lateral migration rates (0.0005 to 0.05 channel widths per year), we found that $T_{cut}$ is proportional
to the inverse of the maximum lateral migration rate such that

$$T_{cut} = c_1 \times \left( \frac{w}{E_{L,max}} \right) \qquad (3)$$

where $w$ is the channel width (meters), $E_{L,max}$ is the maximum lateral migration rate (meters
year$^{-1}$) and $c_1$ is an empirical constant equal to 13.6 ± 3.3 (Figure 2a). Maximum lateral migration
rates ($E_{L,max}$) are used for this comparison because they are specified in the model runs; average
rates are typically a factor of about 3 lower.

Field compilations show that mean lateral migration rates are positively correlated with sediment
fluxes ($Q_s$; Figure 2; Aalto et al., 2008; Grenfell et al., 2012; Constantine et al., 2014). This correla-
tion may reflect the fact that in order for the river channel to migrate by one channel width, sufficient
sediment must be supplied so that a deposit of equivalent volume is created. Along a meander bend,
the volume of the sediment deposit produced after a river migrates one channel width should be
proportional to the product of the bend wavelength ($\lambda$), channel depth ($h$), and channel width ($w$). In
field data (e.g., Williams, 1986), both $\lambda$ and $h$ are correlated with $w$ such that the deposit volume can
be approximated as being proportional to $w^3$. In Figure 2, the correlation between lateral migration
rates and sediment fluxes (Constantine et al., 2014) is recast in terms two timescales: the time re-
quired to migrate one channel width ($w/E_L$) and the time required to supply a proportional volume
of sediment ($w^3/Q_s$). Taken all together, the correlations depicted in Figure 2a-b imply that relative
$T_{cut}$ values can be determined for field systems through comparison of $w^3/Q_s$. That is, natural rivers
with higher $w^3/Q_s$ appear to migrate more slowly and therefore store sediments for longer.

### 2.3 Model for the number of storage events

Following the generic theory presented in Section 2.1, the sediment transit time is equal to the
sum of independent draws from the storage duration distribution with the number of independent
draws being equal to the total number of storage events (Equation 1). Thus, calculating transit time
distributions (TTDs) from the storage duration distribution requires a model for the total number of
storage events sediment undergoes during riverine transit.

Following previous approaches (e.g, Malmon et al. 2003; Lauer and Parker 2008b; Pizzuto et al.
2014), we defined a characteristic length scale over which eroded sediment particles are transported





before being re-deposited. While particles are transported variable distances (depending for example
on particle size and current velocity), we made the simplifying assumption that the dispersion of
the distribution of transport lengths is small relative to the mean transport length. We defined the
characteristic transport length ($x_{tran}$) by balancing the flux of sediment carried downstream with the
lateral flux of sediment that results from channel migration. In this way, $x_{tran}$ represents the length
of channel required to exchange the entire sediment flux with river deposits via lateral migration.
This definition of $x_{tran}$ is comparable to previous studies (Malmon et al., 2003; Lauer and Parker,
2008b; Pizzuto et al., 2014) and was calculated with the equation:

$$x_{\text{tran}} = \frac{Q_s}{E_L \times h} \tag{4}$$

where $Q_s$ is the volumetric sediment flux (m$^3$ yr$^{-1}$), $E_L$ is the mean lateral migration rate (m yr$^{-1}$),
and $h$ is the channel depth (m). The appropriate number of transport events ($n_x$) can be determined
by taking the nearest integer of the ratio of the total channel length to $x_{tran}$. The sediment transit
time distribution can then be determined by convolving the storage duration distribution with itself
$n_x$ times (Equation 1).

### 2.3.1 Prediction of downstream changes

By relating the number of transport events to a characteristic length scale (Equation 4), our model
predicts an increase in transit times as sediments are transported downstream. This relationship be-
tween channel length and transit time forms the basis of our comparison between model results and
field data as it allows datasets without direct measurements of sediment transit times to be used. Cor-
relations apparent in field data (Figure 2b) imply that variations in $x_{tran}$ should be minimal between
river systems such that variations in channel length are the dominant control on $n_x$ (see Equation
4). Here, we set $x_{tran}$ equal to 100 km, which is close to the mean $x_{tran}$ (109±68 km) calculated
from the data compilation shown in Figure 2c and agrees with a complementary, but independent,
analysis by Pizzuto et al. (2016), which suggests that particles enter and exit storage reservoirs 10
times as they transit along a 1000 km long channel.

The predicted changes in sediment transit times with increasing transport length can be cast in
terms of catchment area by taking advantage of the power-law relationship between channel length
($L$; km) and catchment area ($A$; km$^2$; Hack, 1957). This relationship is useful because catchment
areas have been reported for all field data within our compilation (see Section 2.5). We transformed
modeled channel lengths using the equation:

$$L = 1.4 \times A^{0.5} \tag{5}$$

The value of the exponent selected (0.5) reflects a compromise between small ($< 2 \times 10^5$ km$^2$) and
large ($> 2 \times 10^4$ km$^2$) river catchments, which are characterized by larger (0.6) and smaller (0.47)
exponents, respectively (Mueller, 1973). To model catchments with channel lengths shorter than





$x_{tran}$, we assumed they have a mixture of POC with a zero age and POC aged by one transport event. The relative proportion of aged material is equal to the ratio of the channel length to $x_{tran}$.

### 2.4 Linking sediment transit times to POC ages

#### 2.4.1 End-member case with zero cycling

To link our prediction of sediment transit time distributions (TTDs) to POC ages, we started by assuming that the age distribution of POC is equal to the sediment TTD. This can be conceptualized as a system where sediments from a source area contain an initial amount of POC with a zero age. The POC and sediments from the source area then transit through a floodplain with no subsequent POC oxidation or production. This simplified approach serves as a useful end-member case where POC increases in age as much as allowed by sediment storage. To contrast with this, section 2.4.2 describes an approach that explicitly incorporates the effects of POC cycling into the prediction of POC age distributions.

The radiocarbon content of riverine POC is often used as a tracer of the timescale of POC cycling. Typically, radiocarbon measurements are reported as the fraction modern ($Fm$), which is defined as:

$$Fm = \left(\frac{^{14}C}{^{12}C}_{sample*}\right)/\left(\frac{^{14}C}{^{12}C}_{modern}\right) \tag{6}$$

where the subscript sample* refers to the ratio of $^{14}$C to $^{12}$C in a sample normalized to a fixed $^{13}$C to $^{12}$C ratio ($\delta^{13}$C = -25 ‰) and the subscript modern refers to the $^{14}$C to $^{12}$C ratio of a standard. Unlike calendar ages, $Fm$ mixes linearly, making it appropriate for use in systems where POC is composed of a mixture of components with different ages.

Assuming conservative behavior of POC, the appropriate function to transform the sediment transit time distribution into a distribution of $Fm$ values (Equation 2) is the radioactive decay equation:

$$Fm = e^{-\lambda t} \tag{7}$$

where $\lambda$ is the $^{14}$C decay constant ($1.201 \times 10^{-4}$ year$^{-1}$). Here, we assumed that the $^{14}$C/$^{12}$C ratio of the atmosphere is constant in time in order to simplify the model and focus on the role of sediment transport dynamics in setting the radiocarbon content of riverine POC.

#### 2.4.2 Modeling POC cycling in floodplains

If POC is produced and/or consumed during floodplain storage, its age will not be exactly equal to the age of the sediment deposit in which it occurs. If, for example, new POC is produced as older POC is consumed, then the radiocarbon content of bulk POC will increase and be shifted to younger ages relative to the sediment deposit. Production and consumption of POC during sediment storage is consistent with existing radiocarbon measurements from soil chronosequences. Importantly,




soil chronosequence studies from environments where the parent material contains little to no pet-rogenic POC show a general decrease in the $Fm$ of biospheric POC with deposit age (Torn et al., 1997; Lawrence et al., 2015). This relationship implies that, even with active POC cycling, sediment storage will affect the age distribution of riverine POC.

Including the effects POC cycling in our modeling framework requires a description of the kinetics of POC production and consumption. We adopted a simple approach in order to demonstrate the general effects of POC cycling (based on Jenny et al., 1949). This model assumes that the time rate of change in the POC content of a sediment deposit depends on the balance between POC production and consumption. For simplicity, POC consumption is assumed to be first-order with respect to POC

concentrations, which yields the equation

$$\frac{dC}{dt} = P - k(C) \tag{8}$$

where $C$ is the POC concentrations (g cm$^{-3}$), $P$ is the production rate (g cm$^{-3}$ yr$^{-1}$), and $k$ is the consumption rate constant (yr$^{-1}$). This equation predicts that the concentration of POC within a sediment deposit increases with time until it reaches a steady-state concentration equal to $P/k$. The

$e$-folding time of this increase is equal to $1/k$.

To incorporate radiocarbon (as $Fm$) into this model, it is necessary to write separate versions of Equation 8 for $^{12}$C and $^{14}$C. For $^{14}$C, an additional term is required to account for radioactive decay, which yields the equation:

$$\frac{d^{14}C}{dt} = ^{14}P - k(^{14}C) - \lambda(^{14}C) \tag{9}$$

When combined with Equation 8, Equation 9 predicts that the $Fm$ of biospheric POC decreases with time to a steady-state value equal to $(k \times R)/(\lambda + k)$ where $R$ is the ratio of the production rate of $^{14}$C to the production rate of $^{12}$C. For constant steady-state concentrations of POC, the time required for the system to reach steady-state with respect to $Fm$ scales negatively with the POC consumption rate constant (i.e., systems with more slowly cycling carbon require more time to reach

a steady-state).

Equations 8 and 9 can be integrated from the minimum to maximum sediment transit time in order to simulate the evolution of POC concentrations (Equation 8) and $Fm$ (ratio of Equations 8 and 9) with time for different POC production and consumption rates. These integrated equations can be used to transform the storage duration distribution into a distribution of POC concentrations

and $Fm$ for one transport event (Equation 2). To model additional transport events, the means of the concentration and $Fm$ (weighted by concentration) distributions from the preceding transport event were set as the initial values for the integrated forms of Equations 8 and 9 used to transform the storage duration distribution. We used this method to account for in-channel mixing, which will tend to homogenize POC in between transport events. Note that the $Fm$ values modeled in this

manner are not strictly "ages" in any meaningful way, but instead represent the $Fm$ that results from a dynamic balance between POC production, consumption, and radioactive decay.





### 2.4.3 Accounting for the chemical heterogeneity of POC

Applying Equations 8 and 9 requires specifying values for the POC production rate and consumption rate constant. Natural OC is a compositionally heterogeneous material, and cannot be described by

single values for these parameters due to differing rates of biological production and/or resistance to (bio)degradation. To account for this expected heterogeneity, Equations 8 and 9 can sum across multiple POC "pools" with differing production rates ($P$) and consumption ($k$) rate constants (Jørgensen, 1978; Berner, 1980; Boudreau and Ruddick, 1991). For POC concentrations, this can be written as:

$$\frac{dC}{dt} = \sum_{i=1}^{j} P_i - k_i(C_i) \tag{10}$$

where $i$ represents an individual POC pool and $j$ is the total number of POC pools. An analogous equation can be written for $^{14}$C by adding the term for radioactive decay for each POC pool.

Applying Equation 10 requires specifying the number of POC pools as well as their individual steady-state concentrations and consumption rate constants. A simple version of such a model in-

volves two POC pools ($j$ = 2): "fast" cycling POC and "slow" cycling POC. The fast cycling POC pool has a higher steady-state concentration and a higher consumption rate constant relative to the slow cycling pool. As a result, POC concentrations are dominated by the fast pool while $Fm$ values are more sensitive to the slow cycling pool.

Using the sediment TTDs we developed, we calculated the bulk radiocarbon content of riverine

POC using different parameter values of the 2-pool POC cycling model. In all models, the fast cycling POC pool had a fixed consumption rate constant of 0.01 yr$^{-1}$ As long as the rate constant for fast cycling POC is greater than 0.001 yr$^{-1}$, its value has little affect on the model results as this pool cycles rapidly enough to maintain a $Fm \approx 1$. To produce POC with a low $Fm$ value, we set the rate constant describing the slow cycling POC pool to either $2 \times 10^{-6}$ or $2 \times 10^{-5}$ yr$^{-1}$. Production

rates of fast POC were fixed to produce a steady-state concentration of fast POC equal to 0.1 g cm$^{-3}$. Production rates of slow POC were varied such that the steady-state concentration of slow POC was between 5 and 80% of the total (fast + slow) steady-state POC concentration. While consistent with some available field data (e.g., Middelburg, 1989), these parameter values were largely selected in order to produce the range of biospheric $Fm$ values observed in natural rivers (see Section 2.5

below).

### 2.5 Field data compilation and analysis

To benchmark our model results, we compiled field data on the radiocarbon content of riverine POC. As previously mentioned, the bulk radiocarbon content of riverine POC is strongly affected by the mixing of biospheric POC ($Fm \leq 1$) with petrogenic POC ($Fm$=0). Since our model predicts

only the change in the radiocarbon content of biospheric POC, it is necessary that we correct for the




proportion of petrogenic POC in field data. This is accomplished using a modified version of the Galy et al. (2008) two-component mixing model, which requires datasets with more than 2 measurements of $Fm$ per site.

We identified 50 river systems with more than 2 measurements of the concentration and radiocarbon content of riverine POC (full reference list in Table 1). Following Galy et al. (2008), the bulk $Fm$ measurement can be related to the proportions of petrogenic POC ($POC_p$) and biospheric POC ($POC_b$) by the equation:

$$Fm_{bulk} \times [POC]_{bulk} = (Fm_b \times [POC]_b) + (Fm_p \times [POC]_p) \tag{11}$$

where [POC] is the concentration of POC in units of g C g$^{-1}$ sediment. If it is assumed that petrogenic POC is present at a fixed concentration in sediments, then the relationship:

$$[POC]_p = [POC]_{bulk} - [POC]_b \tag{12}$$

can be substituted into Equation 11. Since the $Fm$ of petrogenic C is equal to zero, the assumptions stated above yield the hyperbolic equation:

$$Fm_{bulk} = \frac{Fm_b \times ([POC]_{bulk} - [POC]_p)}{[POC]_{bulk}} \tag{13}$$

All the data from each individual site in our compilation were fit with the non-linear form of the mixing equation (Equation 13) using the Trust-Region algorithm available in the MATLAB 2015a Curve Fitting Toolbox. Since Equation 13 predicts a hyperbolic relationship between POC concentrations and $Fm$ that is concave down, we screened all of the regression results in order to identify instances where a two-component mixing model was inconsistent with the data. Of the 50 river systems in the data compilation, 21 were consistent with the two-component mixing model (Equation 13) and yielded an estimate of the $Fm$ of bulk biospheric POC (Results shown in Table 2). Of these 21 river systems, we excluded one (The Rhône River; biospheric $Fm = 1.41 \pm 0.15$) from further analysis due to its high $Fm$, which we attribute to anthropogenic contamination.

## 3 Results

### 3.1 Meandering model predictions of storage durations

Averaged across our model simulations, the probability distributions of storage durations show a power-law decay in probability as the storage duration increases (Figure 3a). We considered this to be a key feature of the meandering model results, and aimed to capture it in our statistical representation of the modeled age distributions. Without an upper bound, power law distributions can have infinite moments and thus have limited value in describing the full range of behavior of natural systems. Since river systems have a finite size and are expected to eventually recycle more or less all the sediment they store, we employed a tempered Pareto distribution (Cartea and Del-Castillo-Negrete,





2007; Rosiński, 2007) to describe our model results. The tempered Pareto distribution displays a power-law decay until some upper limit where it becomes exponentially tempered. Relative to the truncated Pareto distribution (Mantegna and Stanley, 1994), which simply has a fixed cutoff at some upper limit, the tempered Pareto distribution is less restrictive as it allows the upper bound to also be a stochastic quantity.

Using the approach outlined in Meerschaert et al. (2010), we fit the dimensionless storage duration distribution determined from the meandering model simulations to a tempered Pareto distribution ($p(t)$; Figure 3a), which has a probability density function (pdf) given by:

$$p(t) = \gamma t^{-\alpha-1} \times e^{-t/\beta} \times (\alpha + t/\beta) \tag{14}$$

where $t$ is the storage duration, $\gamma$ is a scale parameter, $\alpha$ is a tail-index, and $\beta$ is a tempering parameter. The $\gamma$ parameter relates to the lower bound of the probability distribution and thus sets the minimum storage time. The $\alpha$ parameter describes the power-law decay in the relationship between probability and storage duration. The $\beta$ parameter describes the storage duration at which power-law behavior ceases and the storage duration distribution begins to follow an exponential function. Our best-fit values of $\gamma$, $\alpha$, and $\beta$ are 1.2, 0.8, and 120. We used equation 14 with these values as the storage duration distribution in order to calculate the sediment TTDs used in all subsequent results (Equation 1).

At steady-state, the storage duration distribution can be uniquely related to the ages of sediment deposits remaining in storage after one transport event (Bolin and Rodhe, 1973; Bradley and Tucker, 2013). Specifically, Bradley and Tucker (2013) demonstrated that the pdf of deposit ages ($p_A(t)$) is proportional to the survivor function (or complementary cumulative distribution function) of storage durations ($S_S(t)$)

$$p_A(t) = \tau \times S_S(t) \tag{15}$$

where $\tau$ is a constant of proportionality that is equal to the ratio of the input/output fluxes to the total reservoir size. Given the relationship between density and survivor functions, Equations 14 and 15 make a prediction for the relationship between the tail-indices of the of the deposit age ($\alpha_d$) and storage duration ($\alpha_s$) distributions. Specifically, $\alpha_d = 1-\alpha_s$ at steady-state. Individually fitting the results of each of our five replicate simulations yields $\alpha_s$ values that range from 0.8 to 1. This range of $\alpha_s$ values, which we assume represents the uncertainty of our estimate, predicts a range of $\alpha_d$ values that overlaps with our best-fit estimate of $\alpha_d$ from fitting Equation 14 to the deposit age distribution derived from fifty replicate model simulations ($\alpha_d = 0.1$; Figure 3a). This consistency between the storage duration and deposit age distributions suggests that our model simulations were at steady-state with respect to sediment storage.

By using the tempered-Pareto distribution (Equation 14) to represent the distribution of storage durations, we excluded storage durations less than the lower bound, which is related to the $\gamma$ parameter and approximately equal to one cutoff time. In our model simulations, the proportion of the




age distribution with storage durations less than the lower bound is small (Figure 3a), which implies
that the lower bound imposed by the tempered-Pareto distribution may not significantly effect our
modeled transit time distributions. Similarly, for modeling POC cycling, a fixed lower bound may
be a reasonable approximation since soil chronosequence studies imply that the onset of significant
organic carbon accumulation is lagged relative to the time of sediment deposition (Torn et al., 1997;
Masiello et al., 2004; Lawrence et al., 2015). Consequently, the small proportion of young sediments
not included in our model is not expected to participate significantly in organic carbon cycling.

### 3.1.1 Non-dimensional model behavior

To show the general model behavior, we started by comparing the dimensionless sediment TTDs
generated using Equations 1 and 14 for 1 to 20 transport events (Figure 3b). As expected, our calcu-
lations show that sediment ages increase with the number of transport events. They also show that
the shape of the transit time distribution changes with increasing number of transport events (Figure
3c). This change in shape is due to the central limit theorem, which states that sum of independent
random variables tends towards a normal distribution even when drawn from a distribution that is
not normal. The central limit theorem applies in this case because the tempered-Pareto distribution
has finite moments due to the exponential tempering of the longest storage durations. However, as a
semi-heavy tailed distribution, sums of tempered-Pareto variables take longer to converge to a nor-
mal distribution relative to exponentially-distributed variables. As a result, sediment TTDs skewed
towards older ages are expected despite the mixing effects of multiple transport events (Figure 3b,c).

The dimensionless sediment TTDs show a roughly linear increase in the mean transit time (MTT)
with increasing number of transport events (Figure 3b). Assuming that $x_{tran}$ is relatively constant,
which is consistent with field data (Figure 2c), the number of transport events should increase with
the ratio of the square root of catchment area to $x_{tran}$ (Equation 5; Hack, 1957). This metric for the
relative number of transport events ($\sqrt{A}/x_{tran}$) can be combined with our metric for relative $T_{cut}$
values ($w^3/Q_s$; Section 2.2.1) to produce a metric for the relative transit time of sediments where:

$$\text{Relative Transit Time} = \frac{\sqrt{A}}{x_{tran}} \times \frac{w^3}{Q_s} \tag{16}$$

This simplified metric captures the expected effects of sediment supply (Constantine et al., 2014)
and channel length (Malmon et al., 2003; Lauer and Parker, 2008b; Pizzuto et al., 2014) on the
duration of sediment storage in river deposits and is useful for comparing systems where more direct
measurements of sediment ages are unavailable.

### 3.2 Coupling sediment storage to POC cycling

### 3.2.1 Radiocarbon as a POC storage tracer

The radiocarbon content of biospheric POC is a tracer of the lifetime of POC in surface environ-
ments, which reflects both the rate of POC cycling and the duration of time over which these reac-





tions occur. For a sediment deposit with a single age, the bulk radiocarbon content (as $Fm$) is set by
the POC production rate, the POC consumption rate constant, as well as the deposit age (Equation
9). For river sediments, which are composed of a mixture of variably-aged deposits (Figure 3), it
is expected that the relationship between the bulk $Fm$, POC cycling parameters, and storage time
will depart significantly from the behavior expected for a single-age deposit (Equation 9). The mag-
nitude of the difference between the heterogeneous (sediment TTD) and homogeneous (single-age
deposit) cases depends on the transit time distribution as well as the values of the POC consumption
rate constants. The direction and magnitude of the offset is important to constrain as it underlies the
quantitative interpretation of field data.

In analyzing the model predictions, we start by comparing the predicted relationships between the
mean transit time (MTT) and bulk $Fm$ for heterogeneous systems with a distribution of sediment
transit times versus homogeneous systems with a single transit time. For this analysis, we re-scaled
the dimensionless sediment TTDs shown in Figure 3b using $T_{cut}$ values selected such that the MTT
for each distribution shape varied between $10^3$ and $10^6$ years. This approach is consistent with field
estimates of MTTs, which range from $10^3$ to $5 \times 10^5$ years when measured using sediment budgeting
(Blöthe and Korup, 2013) or radionuclide approaches (Dosseto et al., 2006; Granet et al., 2010;
Wittmann et al., 2015, 2016; Li et al., 2016).

The exact difference in $Fm$ between the heterogeneous and homogeneous cases depends on the
MTT (Figure 4a). When the MTT is long relative to the time it takes OC cycling to reach steady-state,
there is no difference between the $Fm$ predicted by the heterogeneous and homogeneous models.
However, when a significant portion of the sediment TTD consists of transit times less than the time
required to reach steady-state with respect to OC cycling, there are significant differences between
the $Fm$ predicted by the heterogeneous and homogeneous models. For the cases considered here,
these differences can approach 0.1 $Fm$ units (Figure 4a), which is two orders of magnitude greater
than typical analytical uncertainties. Consequently, applying models based on homogeneous systems
(e.g., Equations 8, 9, and 10) to radiocarbon measurements of riverine POC may yield parameter
values that are off by large factors relative to their true values. In particular, applying homogeneous
models to riverine POC is likely to yield apparent OC cycling rates that are fast relative to the "true"
rates due to the significant proportion of sediments with short transit times.

While heterogeneous sediment TTDs lead to quantitatively distinct relationships between sedi-
ment MTTs and POC $Fm$ values, the large differences in parameter values we selected allow the
OC cycling models to be distinguished from one another despite the differences induced by variable
TTD shapes (Figure 4a). When there is only a very small fraction (i.e., 5%) of slow OC, bulk $Fm$
values remain close to 1 at all MTTs (Figure 4a). Larger portions of slow OC result in more variable
$Fm$ values. For systems with identical values of POC consumption rate constants, increasing the
portion of "slow" OC decreases the $Fm$ observed at a given MTT (Figure 4a).





In general, OC cycling models with lower POC consumption rate constants (i.e., slower OC cy-
cling) have lower steady-state $Fm$ values, but take longer to reach steady-state. As a result, for inter-
mediate MTTs, the bulk $Fm$ of riverine POC can have a higher value for systems with slower POC
cycling rates relative to those with faster cycling rates (Figure 4a). This counter intuitive result stems
from mass balance constraints within the model, which require more slowly cycling compounds to
be produced at slower rates if their concentrations are to remain a fraction of the total POC concen-
tration. In other words, for the same MTT, more slowly cycling POC produced at a slower rate can
yield a similar bulk $Fm$ to more rapidly cycling POC produced at a faster rate (Figure 4).

Results show that a single observation of the bulk $Fm$ of riverine POC can yield a non-unique
interpretation of the underlying POC cycling dynamics even if the sediment TTD is known indepen-
dently (Figure 4a). While this is also true when analyzing soil POC, the chronosequence approach
(i.e., analyzing soils of variable, but independently known ages from the same site) can be used to
better distinguish between different models. Our analysis highlights an analogous approach that can
be applied to river systems where bulk $Fm$ values can be compared within a given river system
or between similar river systems with differing sediment transit times. In our results, OC cycling
rates are distinguished by the shape of the relationship between MTT and bulk $Fm$ (Figure 4a). In
principle, samples with variable MTTs can be collected from the same river system by analyzing
a downstream profile of riverine POC over a length greater than the characteristic transport length
scale (Equation 4).

### 3.3 The downstream profile of POC radiocarbon

For fixed values of POC cycling rates, the bulk $Fm$ of riverine biospheric POC depends on the MTT
(Figure 4a), which increases as the number of transport events increases (Figure 3b). Thus, our model
predicts that the bulk $Fm$ of biospheric POC should decrease downstream. The exact shape of this
decrease depends on the POC cycling rates as well as the relationships between channel length and
sediment transit times set by the transport length scale ($x_{tran}$), the number of transport events ($n_x$),
and the cutoff time ($T_{cut}$).

For short channel lengths, all $Fm$-channel length relationships show $Fm$ values near 1 (i.e., mod-
ern POC) due to limited storage (Figure 4b). As channel length increases, each relationship tends
towards the steady-state $Fm$ that is set by the POC consumption rate constants (Figure 4b). For each
curve shown in Figure 4b, the relative increase in channel length required to reach the steady-state
$Fm$ is set by $T_{cut}$ and $x_{tran}$. In other words, low $Fm$ values require slow POC cycling rates, but
these values can only be expressed with sufficient sediment storage, which increases with increasing
channel length. This result illustrates that the downstream profile of the radiocarbon content of bio-
spheric POC contains information regarding the interplay between sediment storage and OC cycling
(Figure 4b).





### 3.3.1 Benchmark with natural systems

The model results shown in Figures 4a illustrate that increasing the proportion of slow cycling POC results in a decrease in $Fm$, whereas decreasing the consumption rate constant can increase $Fm$. With this in mind, we compared our field data to POC cycling models with fixed POC consumption rate constants (fast and slow rates of 0.01 and $2 \times 10^{-5}$ yr$^{-1}$, respectively), but variable proportions of slow POC (10%, 33%, and 80%). These OC cycling models are contrasted with an "inert" model

(Section 2.4), where POC is assumed to have the same age distribution as the sediments. To produce a range of sediment MTTs, we consider simulations with $T_{cut}$ values of 350 and 1000 years.

All field data within our compilation fall within the region defined by our model predictions after taking into account the uncertainties in our estimates of biospheric $Fm$ from field measurements (Figure 5a). Rivers with small catchment areas tend to have high $Fm$ values while rivers with larger

catchments show a wider range of $Fm$ (Figure 5a). This behavior is predicted by our model wherein sufficient sediment storage is required to express differences in $Fm$ values that result from variable POC cycling rates.

Our model predicts that variations in $Fm$ for rivers with large catchment areas should depend on the rates of OC cycling, which we represent as the portion of slow POC. While it is difficult to

independently quantify POC cycling rates at each of these sites, the observed meridional dependence of POC cycling implies that latitude can be used as a proxy for POC cycling rates (Carvalhais et al., 2014). Consistent with this, the larger catchments with high $Fm$ values are all from low latitudes while the larger catchments with low $Fm$ values are from high latitudes (Figure 5a).

Our model predictions of the relationship between catchment area and $Fm$ are sensitive to both

the portion of slow POC as well as $T_{cut}$ (Figure 5a). In an attempt to control for the dependence of $T_{cut}$ on the relationships shown in Figure 5a, we compare $Fm$ values with the relative transit time metric developed in Section 3.1.1 (Equation 16), which accounts for predicted differences in $T_{cut}$ using variations in $w^3/Q_s$. Comparing the relative transit time of each river with the biospheric $Fm$ yields a relationship consistent with our model predictions (Figure 5b). As the relative transit time

increases, biospheric $Fm$ decreases as a result of radioactive decay during storage (Figure 5b). As expected, the exact relationship between relative transit time and biospheric $Fm$ is variable between sites due to the effects of OC cycling (Figure 5b). By analyzing relative transit times (Figure 5b), we showed that the observed variations in the $Fm$ of riverine biospheric POC are not solely due to variations in OC cycling, but also arise from variations in sediment storage.





## 4 Discussion

### 4.1 Behavior of biogeochemical cycles

The storage of aged POC predicted by our model has implications for global biogeochemical cycles. Observations (Torn et al., 1997; Lawrence et al., 2015) and basic OC cycling models (e.g., Jenny et al., 1949) link the age of a soil or sediment deposit to its radiocarbon content and OC inventory. At the scale of river catchments, the amount of time available for OC accumulation and radioactive decay is set by the patterns of channel migration with time. These patterns lead to the preferential erosion of young deposits (Figure 3a). Consequently, old deposits, which have accumulated more OC, are preferentially retained in floodplains (Figure 3a). Thus, the dynamics of lateral channel migration imply that river deposits are a more sophisticated reservoir of OC than would be assumed by the *null* hypothesis of a well-mixed system due to their age structure. Similarly, river dynamics also influence the expression of these storage processes in geochemical tracers (e.g., $Fm$) by setting the relative proportions of variably-aged river deposits sampled by fluvial erosion (Figure 3).

Typically, the low radiocarbon content of riverine POC is assumed to result from the erosion of petrogenic OC from sedimentary rocks (Masiello and Druffel, 2001; Bouchez et al., 2010; Galy et al., 2008; Tao et al., 2015). Our results suggest that biospheric POC stored in river deposits can be another source of riverine POC that is depleted in radiocarbon. While our model predicts the bulk $Fm$ of this stored OC reservoir, this value is associated with the time-averaged behavior of the river system and represents the average of all POC eroded by channel migration. On short timescales, the $Fm$ of biospheric POC sampled by fluvial erosion may deviate from the time-averaged value predicted by our model depending on the ages of fluvial deposits being eroded. Potentially, such short term variations in the $Fm$ of the biospheric end-member may account for the fact many of the rivers in our compilation do not conform to two-component mixing between petrogenic and biospheric POC (Section 2.5).

Many previous studies of the terrestrial OC cycle have focused on the transport of terrestrial OC to marine depo-centers, where it can be buried and stored over geologic timescales (Galy et al., 2015; Hilton, 2016). While this is undoubtedly an important process, our calculations suggest that sediment storage times in river deposits may approach geologic timescales (e.g., up to $10^6$ years; see also Pizzuto et al., 2016), and thus may play an important role in buffering changes in atmospheric $CO_2$ and $O_2$ concentrations. Moreover, the fluvial processes that dictate sediment storage timescales will also influence how terrestrial OC is transfered to marine basins. As a result, linkages between tectonic/climatic forcings and OC burial fluxes should include the mechanics of the reservoirs associated with terrestrial sediment-routing systems.

Presently, our model only accounts for lateral movements of the river channel with time. In natural river systems, vertical changes driven by overbank deposition, subsidence, and aggradation may also influence the age structure of riverine POC. Since most overbank deposition is focused near



the active channel (Pizutto, 1987; Marriott, 1992; Aalto et al., 2008), the spatio-temporal pattern of lateral channel migration likely approximates the storage durations associated with these deposits. Overbank deposition distal to the channel coupled with subsidence can lead to the burial of sediment deposits below the scour depth of the migrating channel. By selectively removing the oldest deposits,

this additional sediment (and POC) sink can truncate the distribution of sediment ages. Assuming a constant floodplain elevation, the time to bury a deposit beneath the scour depth can be approximated as the ratio of the channel depth to the subsidence rate. As long as this burial timescale is long relative to the upper bound of the storage duration distribution ($\beta$; Equation 14), then subsidence will play a subordinate role in setting the age distributions of riverine sediments and POC. This approach

of comparing the timescales associated with lateral migration and sediment burial can be used to evaluate whether the model presented here is appropriate for a particular field system.

All together, the results of this study imply that the controls on the terrestrial OC cycle are not limited to the factors that affect rates of primary productivity and respiration, but also include the dynamics of terrestrial sedimentary systems (see also: Stallard, 1998; Galy et al., 2015; Hilton, 2016).

Sediment transport processes dictate the time and space scales over which OC cycling occurs, warranting their explicit consideration in models of global biogeochemical cycles. Since the terrestrial biosphere, in turn, influences the behavior of fluvial systems (Tal and Paola, 2007; Gibling and Davies, 2012), there is scope for feedbacks between biogeochemistry, sediment transport processes, and the architecture of fluvial processes to have changed over Earth's history (Algeo and Scheckler,

1998; Gibling and Davies, 2012).

### 4.2 Interpretation of time-varying environmental signals

In addition to affecting the behavior of biogeochemical cycles, our predicted age distributions also have implications for how we interpret environmental changes preserved in sedimentary archives. For example, compound-specific isotopic analyses of terrestrial biomarkers are widely employed as

proxies for environmental conditions (Eglinton and Eglinton, 2008). In systems where these compounds are delivered to a basin via an alluvial river system, they are likely to inherit some age structure as a result of sediment storage. This mixing of variably-aged biomarkers will act as a signal filter and modify the sedimentary expression of environmental changes preserved in the isotopic composition of these compounds (Douglas et al., 2014). While such "shredding" of environmen-

tal signals has been extensively considered for sediments (Jerolmack and Paola, 2010; Ganti et al., 2014; Pizzuto et al., 2016), here, we extended this type of analysis to organic biomarkers.

To explore the implications of our model predictions for the time-series analysis of terrestrial biomarkers, we convolved each simulated sediment transit time distribution with a known periodic signal to investigate the extent of amplitude damping and phase lag at different frequencies. We

specifically looked at the sediment transit time distribution (Figure 3), and not the POC age distribution, to determine the maximum effect of sedimentary averaging on the modulation of environmental



signals. This might also be more appropriate for biomarker studies, which often target recalcitrant compounds that resist degradation (Eglinton and Eglinton, 2008). By treating the problem in this manner, we made the limiting assumption that biomarkers are not continually produced during sed-
iment transit (c.f. Ponton et al., 2014). As such, the results can be viewed as reflecting how the expression of a signal recorded in biomarkers sourced from an upland region is modulated by episodic downstream transport.

    Our results show that low frequency (i.e., forcing period >> mean transit time) environmental signals are likely to be robustly recorded in the isotopic composition of terrestrial biomarkers (Figure
6). However, higher frequency variability shows complex phasing and significant amplitude damping (Figure 6). In part, the large phase lags are due to our limiting assumption that biomarkers are not continually produced during sediment transit. Consequently, the time for a signal generated upstream to advect downstream increases with the number of transport events. As an end-member case, we can consider a scenario where the upland-sourced biomarkers are mixed with an *in situ* floodplain
source that records environmental signals approximately in phase. In this case, the large phase lag between the two biomarker pools will cause destructive interference in the sedimentary expression of the environmental signal.

    As we considered only a single end-member case here, we acknowledge that broad application of our model framework will likely require the explicit modeling of the production and consumption of
organic biomarkers during fluvial transit (Galy et al., 2011; Ponton et al., 2014). Nevertheless, our main point is that knowledge of both the sediment transit time and POC age distributions may aid in the interpretation of proxy systems affected by fluvial averaging. Conversely, if the magnitude of the expected environmental signal was known independently, then its expression within a biomarker record could be used to infer properties of POC age distribution, as is done for water transit through
catchment systems (e.g., McGuire and McDonnell, 2006; Kirchner, 2015). While this general approach was attempted by Douglas et al. (2014) using time-series of biomarkers in lake sediments, they assumed an arbitrary shape for the distribution of POC ages (a bimodal Gaussian distribution). Consequently, our modeling approach can be used to improve such efforts by providing a framework for generating more mechanistic POC age distributions.

**5  Conclusions**

Using simplified models that capture the physical processes associated with sediment storage for meandering rivers, we found that sediment transit times distributions have power-law behavior, though geometric constraints temper or limit the distribution. Coupling our model for sediment transit time distributions to a simple model of OC cycling yields a full model for the radiocarbon content of
riverine POC that can help interpret field observations. Thus we have to consider sediment storage a major aspect of biogeochemical cycling that introduces a time continuum that runs from annual to



potentially million-year timescales. A basic inference from the results of this study is that biomarkers will exhibit a delay and/or mixed signal that convolves both the forcing and the storage. Though complicating the interpretation of sedimentary records, this river-floodplain exchange behavior presents
an opportunity to understand the timescales over which $CO_2$ is stored as organic matter in surface deposits.

## 6  Data Availability

The raw data utilized in the study were compiled from previously published works and are available in the cited manuscripts. In Table 2, we provide results derived from the data compilation. A working
example of our sediment and carbon storage model is included as supplementary MATLAB script.

*Acknowledgements.* M.A.T. acknowledges support from a Caltech Texaco postdoctoral research fellowship, the California Alliance for the Graduate and Professoriate, and the Caltech Discovery Fund. All authors acknowledge the participants of the GE126 course at Caltech (co-taught by W.W.F, M.P.L., and A.J.W.) for providing some of the early ideas that led to this work. Helpful criticism of a draft version of this manuscript was supplied
by Dr. Joel Schiengross.

[Figure 1 about here.]

[Figure 2 about here.]

[Figure 3 about here.]

[Figure 4 about here.]

[Figure 5 about here.]

[Figure 6 about here.]

[Table 1 about here.]

[Table 2 about here.]





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





**List of Figures**







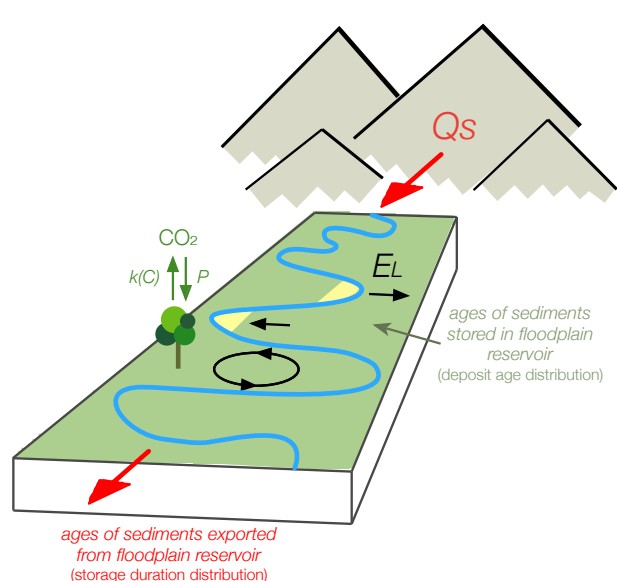

**Figure 1.** Model schematic highlighting key geometric relationships. The sediment flux ($Q_s$) from an upstream area is routed though an alluvial valley. As a result of lateral channel migration ($E_L$), sediment deposits are created and eroded leading to transient sediment storage. During sediment storage, the fixation of atmospheric $CO_2$ by biota leads to the production ($P$) of particulate organic carbon ($C$), which is degraded back to $CO_2$ at a rate ($k$) proportional to its concentration in sediments. At any time, the ages of sediments and organic carbon can be described by two distributions. The storage duration distribution describes the ages of material being actively removed from the system by erosion and thus have reached their maximum age. The deposit age distribution describes the ages of material that remains stored in river deposits and will continue to age until subsequent erosion.





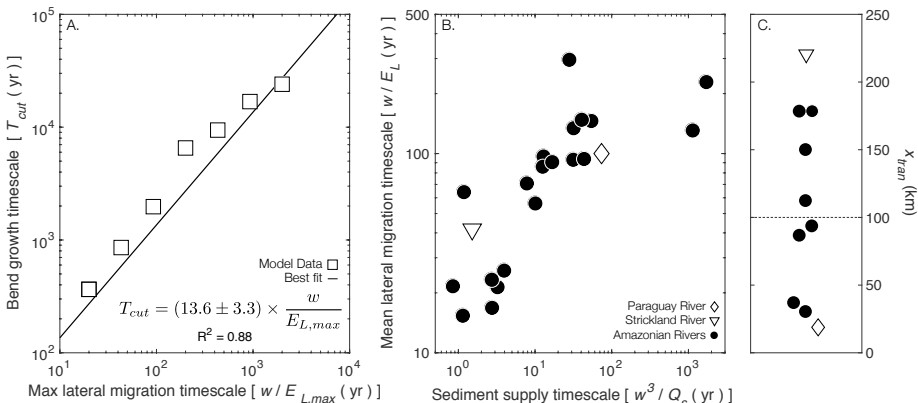

**Figure 2.** Sediment transport parameters in the model and field data. (a) The results of the numerical meandering model simulations that show that the meander bend growth timescale ($T_{cut}$) is correlated with the time required for the channel to migrate one channel width at the maximum lateral migration rate ($E_{L,max}$). (b) Field evidence for a correlation between the mean time required to migrate one channel width ($w/E_L$) and the sediment supply timescale ($w^3/Q_s$). (c) Jitter plot of calculated $x_{tran}$ values (Equation 4) for the field data shown in (b). Together, these panels show that $T_{cut}$ scales with the lateral migration rate, which, in field data, is correlated with suspended sediment fluxes (Constantine et al., 2014). The correlation between lateral migration rates and sediment fluxes results in minimal variation in the transport length scale (c) and provides a proxy for $T_{cut}$ (i.e., $w^3/Q_s$) for rivers where lateral migration rates are unknown.





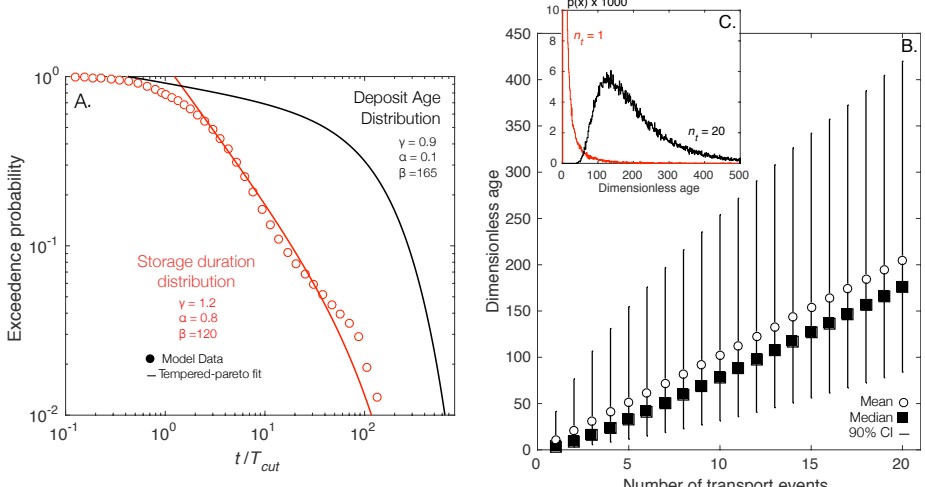

**Figure 3.** Dimensionless age distributions. (a) The survival function (complementary cumulative distribution function) of sediment storage durations (red points) derived from the meandering model. The sediment storage durations are normalized by a cutoff time ($T_{cut}$) of 350 years based on explicit tracking of meander bend growth (Schwenk et al., 2015). The model results were fit to a tempered Pareto distribution and the best-fitting model is shown as a red line. Also shown is a tempered Pareto fit to the distribution of deposit ages at the end of the model simulation (black line). (b-c) The change in the shape of the sediment transit time distribution (normalized by $T_{cut}$; Equation 3) with increasing number of transport events. (b) The mean, median, and 90% confidence interval of the sediment transit time distributions for 1 to 20 transport events. (c) The probability density function of sediment transit time distributions for 1 and 20 transport events. Consistent with the central limit theorem, the shape of the sediment transit distribution varies with the number of transport events.


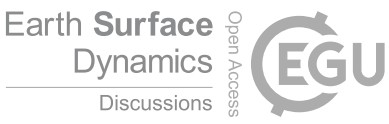

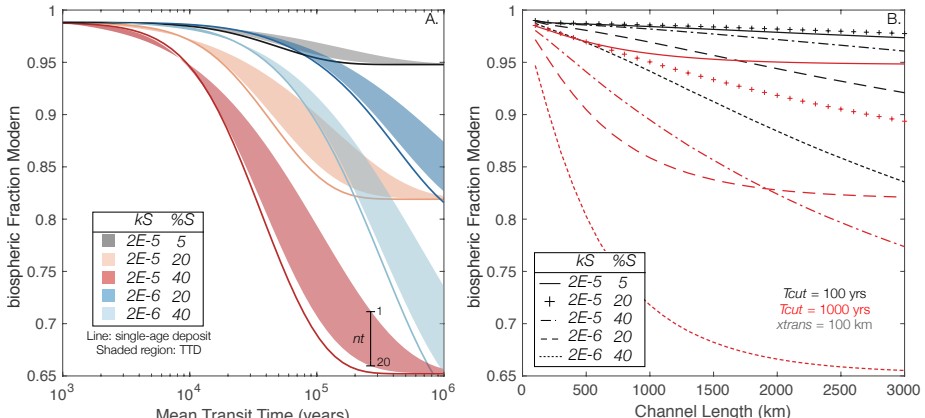

**Figure 4.** Model predictions of the radiocarbon content of POC. (a) Predicted relationships between sediment mean transit times (MTTs) and the $Fm$ of biospheric POC for POC cycling models with variable proportions of slow POC ($\%S$) and variable slow POC consumption rate constants ($kS$). The $Fm$ values shown as shaded areas were calculated using sediment transit time distributions (TTDs) with shapes set by between 1 to 20 transport events (Figure 3) and MTTs between $10^3$ and $10^6$ years. The model results show that plausible sediment TTDs yield significantly different relationships between the MTT and $Fm$ relative to predictions for systems with a single transit time (solid bold lines). While TTD shape is an important factor reflected in the vertical range of each shaded area, the model results suggest that large differences of OC cycling (different colors; see figure legend) are distinguishable in the relationships between the MTT and $Fm$. (b) Predicted downstream profiles of $Fm$ for a fixed $x_{tran}$ (100 km), but variable $T_{cut}$ (100 and 1000 years) and POC cycling parameters (different line styles; see figure legend). The model results suggest that observed downstream profiles are sensitive to both sediment storage timescales, which are controlled by $T_{cut}$, as well as OC cycling parameters.





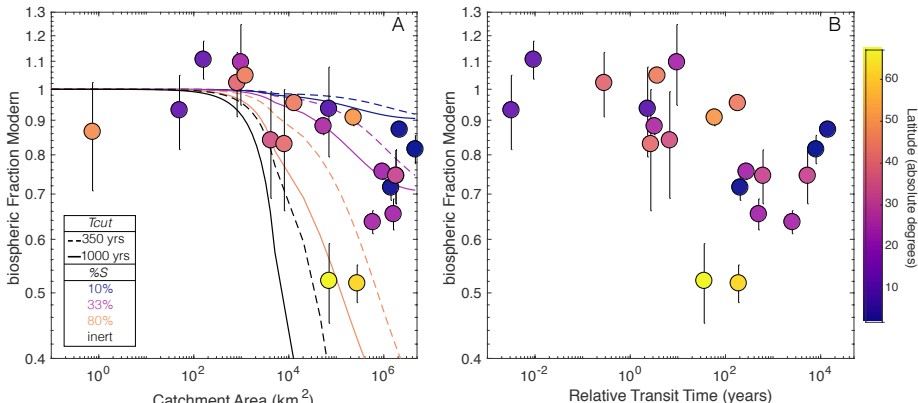

**Figure 5.** Field data compilation and comparison with model predictions. Field data are color coded based on their latitude as a proxy for POC cycling rates. (a) Model curves and field measurements of the relationship between catchment area and bulk biospheric $Fm$. Model curves are drawn for fixed POC consumption rate constants, but variable steady-state proportions of "slow" cycling POC and $T_{cut}$ values (see legend). (b) Measured biospheric $Fm$ values compared to our relative transit time metric (Equation 16), which is based on Hack's Law (Hack, 1957) and scaling relationships shown in Figures 2 and 3. The field data are consistent with the model predictions of aged POC in rivers with larger catchment areas and slower migration rates (inferred from variations in $w^3/Q_s$) and thus longer sediment storage times. We note that in our compilation of rivers with measurements of biospheric $Fm$, catchment area and $w^3/Q_s$ are correlated.





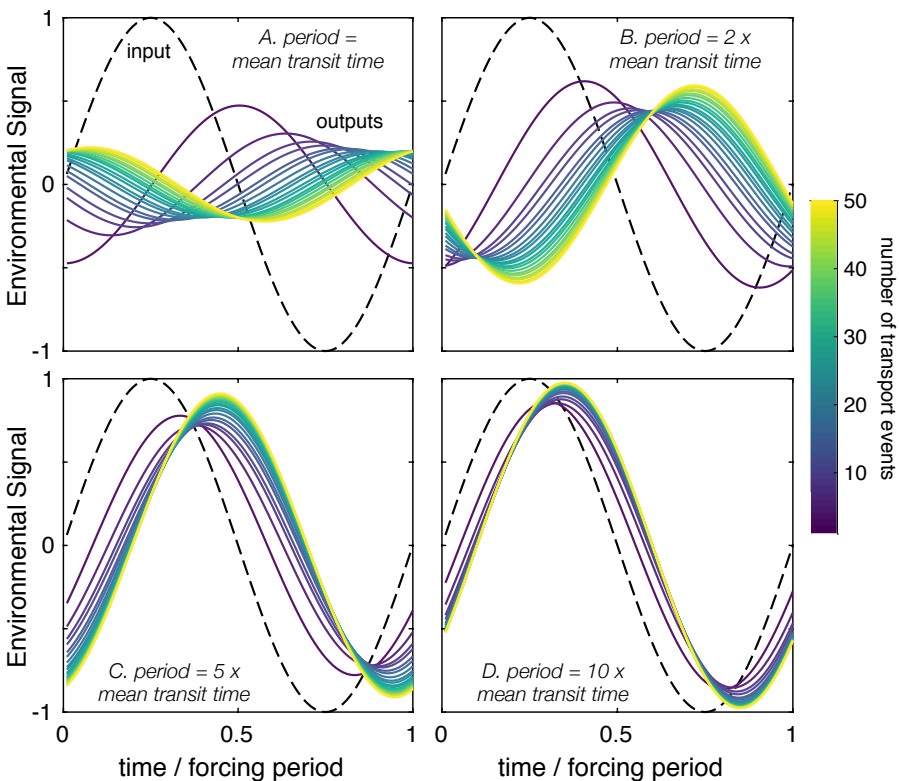

**Figure 6.** Effect of floodplain storage on environmental signals recorded by organic biomarkers. This figure shows the effect of convolving our sediment transit time distributions with a sine wave where the period of the sinusoidal input is equal to the mean transit time (a), twice the mean transit time (b), five times the mean transit time (c), and 10 times the mean transit time (d). The amplitude and phase of the sinusoidal input is shown as the black dashed lines. The colored lines refer to the output signal with the color reflecting the number of transport events (see color bar), which sets the shape of the transit time distribution. The model results suggest that climate signals recorded in terrestrial biomarkers will be significantly modulated if their period is short relative to the mean sediment transit time.





**List of Tables**



**Table 1.** List of all rivers and their associated citations included in the radiocarbon compilation

| River | Citation |
| --- | --- |
| Alsea | Hatten et al. (2012) |
| Amazon | Bouchez et al. (2010, 2014) |
| Arctic Red | Hilton et al. (2015) |
| Atchafalaya | Gordon and Goni (2003); Rosenheim et al. (2013) |
| Avon | Adams et al. (2015) |
| Beni | Bouchez et al. (2010) |
| Brahmaputra | Galy et al. (2007) |
| Calder River | Adams et al. (2015) |
| Changjiang (Yangtze) | Wang et al. (2012); Li et al. (2015) |
| Chontabamba | Townsend-Small et al. (2007) |
| Chorobamba | Townsend-Small et al. (2007) |
| Conwy | Adams et al. (2015) |
| Dee | Adams et al. (2015) |
| Eel | Leithold et al. (2006) |
| Erlenbach | Smith et al. (2013); Turowski et al. (2016) |
| Esperanza | Townsend-Small et al. (2007) |
| Fly | Alin et al. (2008) |
| Fraser | Voss (2014) |
| Ganges | Galy et al. (2007) |
| Garin | Adams et al. (2015) |
| Hodder | Adams et al. (2015) |
| Huancabamba | Townsend-Small et al. (2007) |
| Kosi | Galy et al. (2007) |
| Kosñipata | Clark et al. (2013) |
| Lanyan Hsi | Kao and Liu (1996) |
| Liard | Hilton et al. (2015) |
| Liwu | Hilton et al. (2010) |
| Llamaquiz | Townsend-Small et al. (2007) |
| Madiera | Bouchez et al. (2010, 2014) |
| Meghna | Galy et al. (2007) |
| Mekong | Martin et al. (2013) |
| Mississippi | Rosenheim et al. (2013) |
| Narayani | Galy et al. (2007) |
| Navarro | Leithold et al. (2006) |
| Noyo | Leithold et al. (2006) |
| Peel | Hilton et al. (2015) |
| Pozuzo | Townsend-Small et al. (2007) |
| Rhône | Cathalot et al. (2013) |
| Ribble | Adams et al. (2015) |
| Santa Clara | Masiello and Druffel (2001); Komada et al. (2004) |
| Siuslaw | Leithold et al. (2006) |
| Solimoes | Bouchez et al. (2010, 2014) |
| Strickland | Alin et al. (2008) |
| Tokachi | Nagao et al. (2005) |
| Umpqua | Goñi et al. (2013) |
| Waiapu | Leithold et al. (2006) |
| Waipaoa | Leithold et al. (2006) |
| Yellow (Huanghe) | Wang et al. (2012); Tao et al. (2015); Hu et al. (2015) |
| Zengjiang | Gao et al. (2007) |



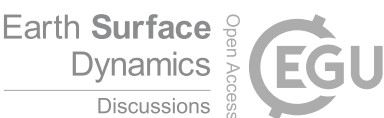

**Table 2.** Physical characteristics and inferred biospheric $Fm$ values for a subset of rivers in the radiocarbon compilation

| River | Sediment Flux (Mt/yr) | Width (m) | Catchment Area (km$^2$) | Biospheric Fm | $\sigma$ | Additional Data Source |
|---|---|---|---|---|---|---|
| Alsea | 0.07 | 65 | 1220 | 1.05 | 0.01 | USGS |
| Amazon | 785 | 4767 | 4618750 | 0.82 | 0.04 | HYBAM |
| Beni | 212 | 410 | 69980 | 0.94 | 0.14 | Wittmann et al. (2015) |
| Brahmaputra | 540 | 4052 | 583000 | 0.64 | 0.03 | |
| Changjiang | 472 | 2000 | 1810000 | 0.75 | 0.07 | Google Earth |
| Eel | 14.00 | 250 | 8063 | 0.83 | 0.17 | USGS |
| Erlenbach | 0.001 | 5 | 0.74 | 0.87 | 0.16 | Nitsche et al. (2012) |
| Fraser | 20 | 450 | 230000 | 0.91 | 0.03 | Google Earth |
| Ganges | 660 | 1910 | 935000 | 0.76 | 0.02 | |
| Kosi | 135 | 412 | 53610 | 0.88 | 0.03 | Google Earth |
| Kosñipata A | 0.06 | 10 | 50 | 0.93 | 0.12 | |
| Kosñipata B | 0.18 | 17 | 160 | 1.11 | 0.07 | |
| Lanyan Hsi | 2.90 | 320 | 980 | 1.10 | 0.15 | Google Earth |
| Liard | 46 | 850 | 275000 | 0.52 | 0.03 | Google Earth |
| Madiera | 433 | 1401 | 1420000 | 0.72 | 0.03 | Google Earth |
| Meghna | 0.24 | 2600 | 1600000 | 0.65 | 0.03 | Google Earth |
| Navarro | 0.56 | 59 | 816 | 1.02 | 0.11 | USGS |
| Peel | 21 | 470 | 70600 | 0.52 | 0.07 | Google Earth |
| Rhône | 31 | 460 | 99000 | 1.41 | 0.15 | Google Earth |
| Santa Clara | 6.84 | 298 | 4200 | 0.84 | 0.15 | USGS |
| Solimoes | 569 | 5893 | 2147740 | 0.87 | 0.02 | HYBAM |
| Umpqua | 1.4 | 435 | 13000 | 0.95 | 0.02 | USGS |