# Peer review of "Model predictions of long-lived storage of organic carbon in river deposits"

_Earth Surface Dynamics, 2017_

## Referee Comment (RC1) · Anonymous Referee #1 · 14 Jun 2017

This manuscript by Torres et al. addresses the role of fluvial sediment transport on the storage and age of river particulate organic carbon (POC). To this aim, it combines an existing numerical model for the evolution of alluvial deposits associated to a meandering river, together with a simplified model for the dynamics of POC in a given sedimentary deposit. Based on these results from numerical simulations, along with considerations on key dimensionless parameters of the model and a compilation of literature data on "biospheric" POC ages, the authors show that river sediment transport dynamics is likely to play a significant role in setting river POC age, and make some predictions on how the interplay between sedimentary and POC dynamics might influence observed biospheric POC ages. This leads to a set of suggestions on how to interpret existing and future field data on river POC in terms of sediment / terrestrial

[Figure]

POC dynamics.

I found this paper very interesting and highly relevant to a large, active research community working on riverine POC and more generally to those using organic sedimentary archives in detrital sediments. It also opens some new perspective for the dynamics of other non-conservative compounds transported by rivers, be setting up an interesting framework. In general I do not have any strong concern regarding the approach nor the conclusions reached by the authors. I have several comments (appended below) mostly regarding missing information or unclear statements, but altogether I recommend publication of this article in Earth Surface Dynamics.

l. 9-10: I find this statement ("sediment transport [. . .] terrestrial realm") a bit weird. Storage in upland soils is clearly also a big player a setting the the maximum time OC persists on land. Although I understand what the authors mean, I do not see why rivers "define" this maximum time, more than upland soils do, for example. I suggest rephrasing.

l. 16 and 77: Doesn't "ameliorates" imply an improvement? If yes, and as I do not see why such "judgmental" word would be needed here, I would rather suggest "modulates", or more simply "affects [too]".

l. 113: Add (if correct) after "transit time distribution": "of this collection of grains".

l. 113-114: I think the physical reason why such transit time distribution is mathematically the result of "n" convolutions of the storage distribution deserves to be more explicitly stated, to keep most readers on board.

Equation 2: First, this equation does not make much sense mathematically speaking. If ˆ14C = f(t) (l. 116-117), in no way can fˆ-1(t) be written (fˆ-1 [provided that it can be defined, which requires the f(t) to be monotonic, by the way, which is not necessarily the case for all C = f(t) functions - depending on POC dynamics - although it is the case in the present paper] is a function of ˆ14C or of something that has the same dimension, at

least). In addition, p_F and p_Tr, as probability distributions, are dimensionless, unlike dt/dˆ14C: eq. (2) therefore has a unit issue. Second, for the sake of clarity, I think the reason why the derivative term appears on the right-hand side should be better explicited, again to allow all the readers to understand what the authors are doing.

l. 147-149: The readers who are not familiar with this model will be interested in knowing how "local and upstream-weighted curvature" influence the local rates of relative channel migration. This is important because later in the paper, it was unclear to me which parameters were directly specified by the authors, and which ones were the result of the model.

l. 154-155: I think the sentence "In the model topology [. . .] bed elevation change" should be moved to l. 147 (after "discrete nodes") as it refers to a very general feature of the model that should be given upfront.

l. 158-175: These explanations would benefit from an example of how a model result looks like, at a given time step or at then of a simulations (this could be added to Fig. 1, the explanatory interest of which is limited). For example, I still wonder whether these are 2D (maps) structures of the river channel and alluvial deposits?

l. 158-188: Reading this, I also wondered how the initial conditions (river channel pathway, initial sediment deposit age distribution. . .) of the alluvial plain were defined in these simulations. This comment relates to some missing information l. 185-188 regarding "replicate simulations": how did these replicate simulations differ exactly (e.g. were the initial conditions randomly set for each simulations, and if yes for which subset of parameters?)? And why a set of replicate simulations for storage duration distribution and another set for deposit age distribution? Why 5 in the first set and 50 in the second? These replicate numbers come up again l. 425 and l. 428, but not anywhere else.

l. 196: The T_cut value of 350 years was obtained from the explicit tracking of meander cut-offs in model runs: is this number actually stable across different replicate

simulations (whatever the difference between these replicate simulations, see comment above)? Exactly equal to 350 years? It seems that T_cut can vary depending on simulation parameters (l. 199-207); so I would guess that this number of 350 years pertains to particular simulation conditions, but this is unclear when reading the article.

l. 216: Add "of" before "two timescales".

l. 217: "E_L" has not be defined at this stage (unlike "E_L,max") - it is defined only l. 238. In addition, the way is retrieved is not clear. I sort of understand that E_L,max is specified by the operator, but E_L is measured and is a mean of all values obtained from the model nodes?

l. 218: To me, the fact that "relative T_cut values can be determined [. . .] through comparison of w^3/Q_s" is implied by the correlation of Fig. 2b, not by the two correlations of Figs. 2a-b.

l. 242: "n_x" is not defined anywhere. I guess it is equal to L / x_tran? If yes, such equation should be added (which requires defining L, defined for now not before l. 256).

l. 287: Avoid using "lambda" (rate constant for radioactive decay of 14C) here to avoid confusion with "lambda" (wavelength of meander bends) l. 213.

l. 309-310: The concentration of POC does not necessarily have to "increase" with time in a sediment deposit (it will decrease if the initial POC concentration is higher than the steady-state concentration).

l. 316: I think the steady-state value of "F_m" is actually (k*R)/(lambda+k)/(14C/12C)_modern (note the division by "(14C/12C)_modern" compared to what is stated l. 316). (k*R)/(lambda+k) is rather the steady-state value of "(14C/12C)_sample*" (following eq. 6).

l. 317: Wouldn't "a given" be more appropriate than "constant" (as "steady state" implies "constant" by definition)?

l. 325: Note that eq. 2 refers only to 14C dynamics - POC and Fm also need an equation for 12C. For this sentence to be correct, eq. 2 would need to be generalized (using e.g. $\hat{n}C = f_n(t)$).

l. 345-347: The relative steady-state values of the fast and slow-cycling POC pools also depend on the production rate P (l. 309 and 316).

l. 351: Add a dot after "0.01 yr$\hat{}$-1".

l. 352: "affect" -> "effect".

l. 397: I'm not arguing against the use of a distribution law with finite moments, but in a way many river systems will not recycle all the sediments they store (e.g. in case of floodplain subsidence, as acknowledged by the authors l. 603-616), or at least only over time scales which make the present model pretty irrelevant. Maybe this statement should be altered to reflect this fact.

l. 421-422: Isn't this ratio of input to output fluxes equal to 1 at "steady-state" (l. 415)? Or does eq. 15 outside of steady-state (unlike what is suggested by the way this whole paragraph is written)? And what is this "reservoir" in the "total reservoir size"?

l. 519-522: I understand that this should be possible "in principle", but in practice this requires that the parameters relevant to the POC dynamics ($k\_S$, %S. . .) remain the same across the river course. This is an important requirement, which might not be fulfilled in many, relatively large river systems.

l. 603-616: This paragraph points out several limitations of the model, which has to be credited to the authors. However, reading this, I thought that it could be slightly extended to reflect the fact that the model used focuses on only one type of river morphology, namely meandering rivers. Although I agree that this river type is widespread, other types of river morphology exist (and are especially represented in the dataset shown in Table 2: braided rivers in the Ganges-Brahmaputra, straight or only slightly bended rivers with stabilized banks by persistent vegetation in the Amazin. . .). While

the authors acknowledge the limitations of their model in terms of processes that might take place even in meandering contexts (such as overbook deposition), this reference to other river morphologies (in which these other processes might be even more important) is lacking. Additionally, one could also emphasize that the sediment grain dynamics addressed by their model referring to banks subjected to erosion / deposition along the channel, the corresponding results are most likely most relevant for coarse grain sizes. However, the "bulk" POC characteristics measured in a river sediment - especially if this sediment is transported as suspension - might be more reflective of fine grains (often OC-richer) that are affected by other processes such as overbook deposition. This could result in patterns for POC characteristics partly decoupled from meandering dynamics.

l. 627-669: Compared to the rest of the article, I find this section a bit weak. First, these "findings" are not related to POC dynamics (especially to processes affecting POC along the lowland river course) at all but rather to grain dynamics, right? And as such they also apply to any tracer deemed as conservative, the signal of which is set in upland rivers and not modified in alluvial plains, right? So why focus the message on biomarkers (which is organic carbon, making the whole point a bit misleading...)? And anyway, aren't these "findings" (Fig. 6) a bit textbook? I mean that most readers will probably know that convoluting a periodic signal with some filter that has a reasonable frequency distribution, the original signal will be dampened and offset in phase, with strongest modulations obtained for high frequency (compared to some representative metric of the frequency of the filter)? Therefore, I suggest removing this section and Fig. 6.

Fig. 3: In panel a, why are the numerical survivor function data (and the corresponding fit) in red on the graph and in black in the legend? Also, It would be informative to represent other functional forms for the possible fit to the numerical data (e.g. exponential...). Finally, The red curve should continue as a flat line below $t/T\_cut \sim$ 1.2, at a value of 1. This is not visible on this figure.

Fig. 4 caption: I think %S is defined for steady state. Maybe this should be written explicitly.

[Figure]

---

## Referee Comment (RC2) · J. Pizzuto (Referee) · 5 Jul 2017

This is a very interesting paper, an initial effort to determine the influence of storage on carbon dynamics of large river systems. There is little doubt that the approach is perhaps oversimplified, and many of the model parameters are poorly constrained, but this is a thought-provoking initial analysis of a neglected and potentially important problem. The manuscript is generally well-written and clearly presented, and the methods and results reasonably convincing and easy to follow.

Jim Pizzuto Dept. of Geological Sciences University of Delaware

Some specific comments, keyed to the text:

1. Line 102 - Please indicate here that you are establishing the usage of an important

term. "Transit time" usually refers to the time spent in a "reservoir", which here would represent time spent "waiting" in an alluvial deposit. Here, however, the term is used to describe how long it takes for a particle to traverse a specified distance along a river corridor. I prefer the term "delivery time" for this concept, but whatever term is used should be defined clearly when introduced to avoid confusion. 2. Line 119, equation (2). Please explain where the dt/dC14 term comes from. 3. Line 245. What does this mean? Transit time usually refers to the time spent in a reservoir. So why do they increase downstream? Or does this refer to the number of storage events as one moves downstream,? Either way, additional explanation would be desirable. 4. Line 252. The Pizzuto et al. reference was published in 2017, not 2016. 5. Line253. The authors might note that Lauer and Parker quote a much larger range in the number of storage times. Also, Pizzuto et al. (2017) note that x_tran increases with transport distance (scale). Might be worth noting here. 6. Lines 320-330. I didn't really understand the description of the mathematics here. More, and clearer, explanation is needed if readers are expected to really understand what the authors are doing here. 7. Line 355. Values selected for these parameters seem pretty arbitrary and perhaps not too well justified, but...ok. 8. Line 415. Please discuss the assumption of a steady state in the methods section. It is common in reservoir theory modeling but a rather extreme prediction for natural fluvial systems. In the rivers intended for this paper to represent, what is the characteristic time scale for a steady state to be achieved? Is this a reasonable assumption? Likely not. Perhaps this merits some discussion....in the discussion section of the manuscript, as well as in the methods section. 9. Line 563. How is Qs assessed? From stream gaging station records? Are these estimated given in a table somewhere in the manuscript? They should be. More discussion of these data is warranted, also. Generally, useful estimates of Qs are not available. 10. Line 573. "though geometric constraints temper or limit the distribution." This is not DEMONSTRATED in the manuscript, it is really simply assumed. The text should be modified to reflect this – it is not a RESULT obtained either from data analysis or computations, but an assumption of the author's approach. 11. Line 919. Pizzuto's name is misspelled here.

12. Line 927. Correct citation year is 2017, not 2016. 13. Figure 1. Isn't the length of the valley reach an important variable to consider? How about the geometry of the meandering river domain simulated, perhaps in units of river widths or something? Please explain and clarify. It is also possibly worth noting that the storage time distribution as defined here cannot be measured using observations, unless suspended particles in transport could be "tracked" and dated in some way. It is more elegant to determine the ages of particles as they leave a storage reservoir by dating eroding bank deposits, for example. This definition of storage time can actually be defined by field measurements. 14. Figure 2, panel 2. The range of x_tran quoted by Pizzuto et al. 2017 is much larger than the data illustrated here. This should be noted in the manuscript. 15. Figure 3. It is odd to show the storage duration data in red, but then present the legend associated with these data in black. Please keep the color scheme consistent. 16. Figure 6. Great figure!

---

## Author Comment (AC1) · 2 Aug 2017

**Author responses and changes made in revision**

In response to the thorough and thoughtful reviews provided by an anonymous reviewer and Dr. Pizzuto, we have substantially revised the main text of our manuscript and have added a new panel to Figure 1. We feel that these revisions appropriately address all of the comments provided by each of the reviewers and have yielded an overall stronger manuscript. Below, we provide a detailed response to each individual comment from both reviewers along with an explanation of the changes made to the manuscript.

Reviewer #1

l. 9-10: I find this statement ("sediment transport [. . .] terrestrial realm") a bit weird. Storage in upland soils is clearly also a big player a setting the the maximum time OC persists on land. Although I understand what the authors mean, I do not see why rivers "define" this maximum time, more than upland soils do, for example. I suggest rephrasing.

This is a good point. In addition to during fluvial transport and storage, sediments "age" as they are uplifted by tectonic forces and subsequently transported down hillslopes. We have modified our sentence to reflect these additional controls on sediment ages. Line 37 now reads, *"Thus, rivers influence the amount of time POC can persist within the terrestrial realm and integrate over areas that are large compared to the spatial scales of variability in biogeochemical processes."* A similar change has been made to the abstract.

l. 16 and 77: Doesn't "ameliorates" imply an improvement? If yes, and as I do not see why such "judgmental" word would be needed here, I would rather suggest "modulates", or more simply "affects [too]".

We have replaced "ameliorate" with "limit" in these sentences.

l. 113: Add (if correct) after "transit time distribution": "of this collection of grains".

We added this suggested phrase.

l. 113-114: I think the physical reason why such transit time distribution is mathematically the result of "n" convolutions of the storage distribution deserves to be more explicitly stated, to keep most readers on board.

For clarity, we modified this sentence on line 114 to read, *"Mathematically, the transit time distribution of this collection of grains ($p_{Tr}(t)$), which is the distribution function for the sum of n random values drawn from $p_S(t)$, is given by the convolution of the storage distribution by itself n times."*

Equation 2: First, this equation does not make much sense mathematically speaking. If ˆ14C = f(t) (l. 116-117), in no way can f⁻1(t) be written (f⁻1 [provided that it can be defined, which requires the f(t) to be monotonic, by the way, which is not necessarily the case for all C = f(t) functions - depending on POC dynamics - although it is the case in the present paper] is a function of ˆ14C or of something that has the same dimension, at least). In addition, p_F and p_Tr, as probability distributions, are dimensionless, unlike dt/dˆ14C: eq. (2) therefore has a unit issue. Second, for the sake of clarity, I think the reason why the derivative term appears on the

right-hand side should be better explicited, again to allow all the readers to understand what the authors are doing.

Very helpful! We thank the reviewer for catching our notation error. The inverse of the function $^{14}C = f(t)$ is $t = f^{-1}(^{14}C)$. Substituting in this correct version of the inverse equation also fixes the issue with the units by canceling $dt/d^{14}C$.

l. 147-149: The readers who are not familiar with this model will be interested in knowing how "local and upstream-weighted curvature" influence the local rates of relative channel migration. This is important because later in the paper, it was unclear to me which parameters were directly specified by the authors, and which ones were the result of the model.

We completely agree that the details of the numerical meandering model are relevant to the present study, which is why we dedicated multiple paragraphs in Section 2.2 to describing our application of the Howard and Knutson (1984) model. However, it is beyond the scope of our manuscript to re-state all of the model equations. Instead, readers are referred to the original publication and Limaye and Lamb (2013) for additional model details.

To make our model description clearer and address the reviewer's comment, we added the sentence: *"Smoothly curving meander bends initially develop from small perturbations in the channel centerline trace, and migrate at rates that vary in both space and time."* to the model description starting on line 155. This, along with addition of other model details requested by both reviewers, should help readers better understand our application of the Howard and Knutson (1984) model.

l. 154-155: I think the sentence "In the model topology [. . .] bed elevation change" should be moved to l. 147 (after "discrete nodes") as it refers to a very general feature of the model that should be given upfront.

We moved this sentence to the beginning of the paragraph as suggested (now starting on line 152).

l. 158-175: These explanations would benefit from an example of how a model result looks like, at a given time step or at then of a simulations (this could be added to Fig. 1, the explanatory interest of which is limited). For example, I still wonder whether these are 2D (maps) structures of the river channel and alluvial deposits?

Good point. We added a new panel to Figure 1 (panel B) that shows example model output.

l. 158-188: Reading this, I also wondered how the initial conditions (river channel pathway, initial sediment deposit age distribution. . .) of the alluvial plain were defined in these simulations. This comment relates to some missing information l. 185-188 regarding "replicate simulations": how did these replicate simulations differ exactly (e.g. were the initial conditions randomly set for each simulations, and if yes for which subset of parameters?)? And why a set of replicate simulations for storage duration distribution and another set for deposit age distribution? Why 5 in the first set and 50 in the second? These replicate numbers come up again l. 425 and l. 428, but not anywhere else.

Replicate simulations were generated using using different initial channel centerlines generated by adding random perturbations to a straight line. We revised the text to make this clear for the reader. Line 196 now reads, *"Different initial channel centerlines for the replicate model runs were generated by adding random perturbations on the order of 0.01 channel widths to otherwise straight channels. These perturbations also initiate meandering in the model runs"*.

To track the age distribution of sediments being eroded at each time step (storage duration), we needed to modify the code used in Limaye and Lamb (2013) and reprocess the original model results. Therefore, we reprocessed a subset (5) of the original model runs (50) to track storage durations. To make this clearer, the manuscript now states (Line 195), *"We used a subset of five replicate simulations to determine storage duration distributions."*

l. 196: The T_cut value of 350 years was obtained from the explicit tracking of meander cut-offs in model runs: is this number actually stable across different replicate simulations (whatever the difference between these replicate simulations, see comment above)? Exactly equal to 350 years? It seems that T_cut can vary depending on simulation parameters (l. 199-207); so I would guess that this number of 350 years pertains to particular simulation conditions, but this is unclear when reading the article.

This is a valuable point. The $T_{cut}$ of 350 years pertains to the simulations used to generate the age distributions. We did run additional simulations to determine the relationship between maximum channel migration rates and $T_{cut}$, but did not use or report age distributions generated from these additional model runs.

In a single model run, the timescale for meander bend growth and cutoff varies. Based on an ensemble of 3105 cutoffs from an individual model run, we determined that the peak in the probability distribution of cutoff times was equal to 350 years and represented the characteristic $T_{cut}$ for the given set of model parameters. By measuring $T_{cut}$ from a large population of individual cutoff events, it accounts for variations in channel trajectories between replicate model simulations (i.e., model runs with identical model parameters but different initial channel centerlines). In the main text, we have added information pertaining to the calculation and selection of $T_{cut}$ (Line 206), which reads, *"While the cutoff time varied in the model runs, the peak of the probability distribution function occurred for Tcut = 350 years"*.

l. 216: Add "of" before "two timescales".

This additional word has been added.

l. 217: "E_L" has not be defined at this stage (unlike "E_L,max") - it is defined only l. 238. In addition, the way is retrieved is not clear. I sort of understand that E_L,max is specified by the operator, but E_L is measured and is a mean of all values obtained from the model nodes?

We have added a definition of $E_L$ in the text (Line 217). The reviewer is correct in assuming that it is a mean of all values along a river channel.

l. 218: To me, the fact that "relative T_cut values can be determined [. . .] through comparison of wˆ3/Q_s" is implied by the correlation of Fig. 2b, not by the two correlations of Figs. 2a-b.

We cite both figure panels as we feel that correlation between $T_{cut}$ and lateral migration rates

(panel A) is relevant since only migration rates are directly measured in the field.

l. 242: "n_x" is not defined anywhere. I guess it is equal to L / x_tran? If yes, such equation should be added (which requires defining L, defined for now not before l. 256).

Yes, the reviewer is correct in assuming $n_x$ is calculated from $L/x_{tran}$. We have added a definition of L and $n_x$ in the text (Line 252).

l. 287: Avoid using "lambda" (rate constant for radioactive decay of 14C) here to avoid confusion with "lambda" (wavelength of meander bends) l. 213.

Agreed that it is unfortunate that both radioactive decay constants and wavelengths share the lambda symbol, however because both are so ingrained we are reluctant to use an alternative symbol for the $^{14}C$ decay constant as lambda is the standard symbol. However, for clarity, we now use $\lambda$ to represent the $^{14}C$ decay constant and $\lambda_{bend}$ to represent the meander bend wavelength.

l. 309-310: The concentration of POC does not necessarily have to "increase" with time in a sediment deposit (it will decrease if the initial POC concentration is higher than the steady-state concentration).

This is a great point. We have modified the text accordingly. It now reads, *"This equation predicts that the concentration of POC within a sediment deposit evolves towards a steady-state concentration equal to P/k. The e -folding time of this evolution is equal to 1/k ."* (Line 323).

l. 316: I think the steady-state value of "F_m" is actually (k*R)/(lambda+k)/(14C/12C)_modern (note the division by "(14C/12C)_modern" compared to what is stated l. 316). (k*R)/(lambda+k) is rather the steady-state value of "(14C/12C)_sample*" (following eq. 6).

Yes. We have modified the text accordingly. It now reads, *"When combined with Equation 8, Equation 9 predicts that the Fm of biospheric POC decreases with time to a steady-state value equal to k/(λ+k)"* (Line 330).

l. 317: Wouldn't "a given" be more appropriate than "constant" (as "steady state" implies "constant" by definition)?

Yes! We have modified the text accordingly.

l. 325: Note that eq. 2 refers only to 14C dynamics - POC and Fm also need an equation for 12C. For this sentence to be correct, eq. 2 would need to be generalized (using e.g. ˆnC = f_n(t)).

Corrections made. The sentence on line 336 now reads, *"These integrated equations can then be used to transform any age distribution into a distribution of POC concentrations and Fm (analogously to Equation 2)."*

l. 345-347: The relative steady-state values of the fast and slow-cycling POC pools also depend on the production rate P (l. 309 and 316).

Specifying both the steady-state value and the consumption rate constant fixes the production rate.

l. 351: Add a dot after "0.01 yrˆ-1".

Added.

l. 352: "affect" -> "effect".

Changed.

l. 397: I'm not arguing against the use of a distribution law with finite moments, but in a way many river systems will not recycle all the sediments they store (e.g. in case of floodplain subsidence, as acknowledged by the authors l. 603-616), or at least only over time scales which make the present model pretty irrelevant. Maybe this statement should be altered to reflect this fact.

This point is well taken. We have modified our statement on line 418, which now states, *"Since our model simulations show evidence for an upper bound (Figure 3a) and natural river systems have a finite size and, in the absence of external forcing, are expected to eventually recycle more or less all the sediment they store, we employed a tempered Pareto distribution (Cartea and Del-Castillo-Negrete, 2007; Rosiński, 2007) to describe our model results"*.

l. 421-422: Isn't this ratio of input to output fluxes equal to 1 at "steady-state" (l. 415)? Or does eq. 15 outside of steady state (unlike what is suggested by the way this whole paragraph is written)? And what is this "reservoir" in the "total reservoir size"?

Yes, at steady-state, input and output fluxes are equal. So, either can be compared to the total sediment reservoir size to calculate τ. We have modified the text to make this clearer. Line 444 now reads, *"… where τ is a constant of proportionality that is equal to the ratio of the total sediment reservoir size to either the input or output fluxes."*

l. 519-522: I understand that this should be possible "in principle", but in practice this requires that the parameters relevant to the POC dynamics (k_S, %S. . .) remain the same across the river course. This is an important requirement, which might not be fulfilled in many, relatively large river systems.

In our sentence, we stated that sampling sediments along a downstream profile could provide samples with variable mean transit times. This statement is independent of POC dynamics. However, we agree that, when interpreting the radiocarbon content of riverine POC, one has to consider spatial variations in OC cycling rates as well as changes in sediment ages. As our sentence, as written, only pertains to sediment transit times, we have opted not to modify it further. However, we now acknowledge the potential complications of spatial variations in OC cycling on line 548 with the sentence, *"However, spatial variations in OC cycling rates may complicate the interpretation of such analyses"*.

l. 603-616: This paragraph points out several limitations of the model, which has to be credited to the authors. However, reading this, I thought that it could be slightly extended to reflect the

fact that the model used focuses on only one type of river morphology, namely meandering rivers. Although I agree that this river type is widespread, other types of river morphology exist (and are especially represented in the dataset shown in Table 2: braided rivers in the Ganges-Brahmaputra, straight or only slightly bended rivers with stabilized banks by persistent vegetation in the Amazin. . .). While the authors acknowledge the limitations of their model in terms of processes that might take place even in meandering contexts (such as overbook deposition), this reference to other river morphologies (in which these other processes might be even more important) is lacking. Additionally, one could also emphasize that the sediment grain dynamics addressed by their model referring to banks subjected to erosion / deposition along the channel, the corresponding results are most likely most relevant for coarse grain sizes. However, the "bulk" POC characteristics measured in a river sediment - especially if this sediment is transported as suspension - might be more reflective of fine grains (often OC-richer) that are affected by other processes such as overbook deposition. This could result in patterns for POC characteristics partly decoupled from meandering dynamics.

This is a good point because our efforts here are focused on rivers where sedimentation is tied to lateral migration of the channel. While this doesn't capture all alluvial river systems that one might like to examine, lateral channel migration is common to many types of river. So, in that way, our results are generalizable. Similarly, while we neglect overbank deposition, field studies show that it is focused near the channel (Pizzuto 1987, Marriot 1992, Aalto et al. 2008). As a result, the dynamics of lateral channel migration should also affect the storage times of sediments deposited via overbank deposition. These points are both stated in the main text (Starting on lines 138 and 630). On line 630, we modified the text to read, "*Presently, our model only accounts for lateral movements of a meandering river with time. In natural river systems, channel pattern (Eaton et al. 2010) and elevation changes driven by overbank deposition, subsidence, and aggradation may also influence the age structure of riverine POC.*"

l. 627-669: Compared to the rest of the article, I find this section a bit weak. First, these "findings" are not related to POC dynamics (especially to processes affecting POC along the lowland river course) at all but rather to grain dynamics, right? And as such they also apply to any tracer deemed as conservative, the signal of which is set in upland rivers and not modified in alluvial plains, right? So why focus the message on biomarkers (which is organic carbon, making the whole point a bit misleading. . .)? And anyway, aren't these "findings" (Fig. 6) a bit textbook? I mean that most readers will probably know that convoluting a periodic signal with some filter that has a reasonable frequency distribution, the original signal will be dampened and offset in phase, with strongest modulations obtained for high frequency (compared to some representative metric of the frequency of the filter)? Therefore, I suggest removing this section and Fig. 6.

We completely understand this point. It is well known that filtering a periodic signal modifies its phase and amplitude. In this section, we sought to illustrate how the mechanics outlined in our study could be used to help identify the shape of the filter most relevant to terrestrial biomarker studies. In previous work on catchment biomarker storage (e.g., Douglas et al. 2014), researchers selected arbitrary filter shapes. While our study is unlikely to provide the exact shape, we suspect that the heavy-tailed dynamics of river systems are imprinted in the age distribution of terrestrial biomarkers.

In the revised version of our manuscript, we have opted to keep Figure 6, but have rewritten the associated text to better emphasize the points mentioned above and the fact that

amplitude dampening / phase lag is a generic outcome of signal filtering. For example, on Line 668, we state, *"While the extent of amplitude damping and phase lag at different frequencies is a generic outcome of signal filtering (Figure 6), our model framework links these effects to physical properties of river systems (e.g., lateral migration rates, sediment fluxes, and channel length)"*. In part, this decision was made because the other reviewer, Dr. James Pizzuto, had a much more positive view of Figure 6 and the associated analysis.

Fig. 3: In panel a, why are the numerical survivor function data (and the corresponding fit) in red on the graph and in black in the legend? Also, It would be informative to represent other functional forms for the possible fit to the numerical data (e.g. exponential. . .). Finally, The red curve should continue as a flat line below t/T_cut _ 1.2, at a value of 1. This is not visible on this figure.

We apologize for the labeling confusion and have modified the color scheme appropriately. Additionally, we have added an exponential distribution to panel A of the figure for comparison.

Fig. 4 caption: I think %S is defined for steady state. Maybe this should be written explicitly.

Good point. This description of the variable has been added to the figure caption.

Reviewer #2 (Dr. Pizzuto)

1. Line 102 - Please indicate here that you are establishing the usage of an important term. "Transit time" usually refers to the time spent in a "reservoir", which here would represent time spent "waiting" in an alluvial deposit. Here, however, the term is used to describe how long it takes for a particle to traverse a specified distance along a river corridor. I prefer the term "delivery time" for this concept, but whatever term is used should be defined clearly when introduced to avoid confusion.

We agree that the distinction between different age-related variables is crucial and often varies between research groups. Following work in catchment hydrochemistry, we use transit time to refer to the age distribution of material exiting the system, which, for sediments, will be the cumulative time spent "waiting" in alluvial deposits. By defining a characteristic length scale over which sediments are transported before being re-deposited, we link the transport distance and the sediment transit time. However, we do not define the transit time as the time required to travel a specific distance. To make this as clear as possible for the reader in the main text, we have revised the prose to make these definition and distinctions clearer. For example, on line 109 we state, *"Thus, for some number of storage events n, the cumulative age, or transit time, of the sediment can be calculated as the sum of n draws from the probability distribution of storage durations"*.

2. Line 119, equation (2). Please explain where the dt/dC14 term comes from.

This term is necessary to convert a distribution of ages into a distribution of radiocarbon concentrations. After correcting a notation error pointed out by reviewer #1, Equation 2 should now be easier to follow.

3. Line 245. What does this mean? Transit time usually refers to the time spent in a reservoir. So why do they increase downstream? Or does this refer to the number of storage events as

one moves downstream,? Either way, additional explanation would be desirable.

We expect that, with increased transport distance, sediments will enter and exit alluvial storage more times leading to cumulative aging. As a result, the age distribution of material in the channel, which we refer to as the transit time distribution, will increase downstream. We have modified the text to make this point more clearly. Line 256 now reads, *"By relating the number of transport events to a characteristic length scale (Equation 4), our model predicts that sediment transit times will increase with increasing transport distance due to repeated storage in fluvial deposits."*

4. Line 252. The Pizzuto et al. reference was published in 2017, not 2016.

We have corrected this typographical error.

5. Line253. The authors might note that Lauer and Parker quote a much larger range in the number of storage times. Also, Pizzuto et al. (2017) note that x_tran increases with transport distance (scale). Might be worth noting here.

Good point. We revised the prose to note that x_tran may increase with increasing transport distance on line 266 by stating, *"However, we note that Pizzuto et al. (2017) also predict that the transport length scale increases with increasing river catchment area"*.

6. Lines 320-330. I didn't really understand the description of the mathematics here. More, and clearer, explanation is needed if readers are expected to really understand what the authors are doing here.

This is a really helpful point. In this section, we intended to emphasize the logic behind our approach. Directly transforming the transit time distribution using the OC cycling function (as outlined in Equation 2) would imply that, between storage events, a sediment deposit "knows" to start with the same POC concentration and *Fm* it ended with in the previous storage event. In reality, we expect that, between transport events, POC and sediments sourced from multiple variably aged deposits are homogenized in the active river channel. To account for this mixing, we set the initial POC concentration and *Fm* of a sediment deposit to the average values produced from the preceding storage event. Since this only requires changing the initial values for Equations 8 and 9, we elected not to walk the reader through the arithmetic. However, we have edited this entire section for clarity.

The section, which starts on line 335, now reads, *"Equations 8 and 9 can be integrated in order generate equations that predict the concentration and Fm of POC as a function of time. These integrated equations can then be used to transform any age distribution into a distribution of POC concentrations and Fm (analogously to Equation 2). Simply transforming the transit time distribution would imply that, in between transport events, POC sourced from the erosion of variably-aged sediment deposits is not mixed together in the river channel, which is unlikely to be true. Instead, to simulate in-channel mixing, the initial concentration and Fm of POC deposited can be made to vary downstream based on the extent of aging during each transport event.*

*For the first transport event, the storage duration distribution is transformed with the integrated forms of Equations 8 and 9 where the initial POC concentrations are equal to zero. This*

*accounts for the POC produced and consumed during the first storage event as sediment transits through a river system. When these upstream sediment deposits are eroded, they will transport a mixture of variably-aged POC downstream until it is re-deposited. We assume that the initial concentration and Fm of POC incorporated into these downstream deposits is equal to the means of the concentration and Fm (weighted by concentration) of eroded deposits from the preceding storage event. These initial values are used to generate new functions from Equations 8 and 9 to transform the storage duration distribution for the next storage event".*

7. Line 355.Values selected for these parameters seem pretty arbitrary and perhaps not too well justified, but. . .ok.

We readily acknowledge that to an extent these values appear arbitrary because there are few strong empirical constraints. Because of this, we chose values that match the data compilation of Middleburg (1989).

8. Line 415. Please discuss the assumption of a steady state in the methods section. It is common in reservoir theory modeling but a rather extreme prediction for natural fluvial systems. In the rivers intended for this paper to represent, what is the characteristic time scale for a steady state to be achieved? Is this a reasonable assumption? Likely not. Perhaps this merits some discussion. . ..in the discussion section of the manuscript, as well as in the methods section.

This is an important point. While "steady-state" assumptions are often powerful for modeling studies, it's never certain how much value they hold for natural systems. Because of the way that we constructed our model calculations, we do not expect that the simulations do a good job of capturing the processes and timescales associated with dynamic changes in things like sediment fluxes or migration rates.

However, it is also not known by how much the model results would differ between the steady-state and transient case. Consequently, we have not labored to try to extract that type of meaning from our calculations. The point of our approach here is to provide some new context for the interpretation of field measurements. In that sense, our steady-state model is useful in that it identifies measurable parameters that are expected to influence the radiocarbon content of riverine POC. In our compilation of field data, we observed trends consistent with our model predictions, which highlights the utility of a steady-state model despite the expectation that natural systems may not all be at steady-state.

We have now added an additional mention of our assumption of steady state in the methods section on line 178. Specifically, we state, *"Thus, we consider only the time-averaged behavior of the model under steady conditions without any additional forcing."*

9. Line 563. How is Qs assessed? From stream gaging station records? Are these estimated given in a table somewhere in the manuscript? They should be. More discussion of these data is warranted, also. Generally, useful estimates of Qs are not available.

In Table 2, we report the Qs values that we use in the second column, which is labeled sediment flux. Largely, these values are from sediment gauging records. While there are possible biases with sediment gauging measurements, we compared catchments with sediment fluxes that vary by over five orders of magnitude. We expect that these large, firstorder differences between sites are accurate, which is what we focus on in Figure 5b. On line 486, we now state, *"We note that field estimates of sediment fluxes are often imperfect (e.g., Kirchner et al. 2001). However, we expect that large differences in sediment fluxes between field sites, such as those in our data compilation, will provide meaningful insights into differences in sediment storage using Equation 16"*.

10. Line 573. "though geometric constraints temper or limit the distribution." This is not DEMONSTRATED in the manuscript, it is really simply assumed. The text should be modified to reflect this – it is not a RESULT obtained either from data analysis or computations, but an assumption of the author's approach.

This is a fair point, but in addition to basic expectations for real limits on size, our results support this inference. Our implementation of the numerical meandering model does show an inflection point in the probability distribution of storage durations consistent with exponential tempering (Figure 3a). So, in that sense, tempering is not purely an artifact of our decision to fit the model data with a particular probability distribution. This is now emphasized on line 418 where we state, *""Since our model simulations show evidence for an upper bound (Figure 3a) and natural river systems have a finite size and, in the absence of external forcing, are expected to eventually recycle more or less all the sediment they store, we employed a tempered Pareto distribution (Cartea and Del-Castillo-Negrete, 2007; Rosiński, 2007) to describe our model results"*.

However, we do assume that natural systems also show some degree of tempering, which we do not test in this manuscript. We have modified the manuscript to better reflect this assumption.  On line 692, we now state, *"Using simplified models that capture the physical processes associated with sediment storage for meandering rivers, we found that sediment transit times distributions have power-law behavior, though geometric constraints temper or limit the distribution in our model simulations."*

11. Line 919. Pizzuto's name is misspelled here.

We apologize for this typographical error! All corrections have been made to the bibliographic entries.

12. Line 927. Correct citation year is 2017, not 2016.

Fixed.

13. Figure 1. Isn't the length of the valley reach an important variable to consider? How about the geometry of the meandering river domain simulated, perhaps in units of river widths or something? Please explain and clarify. It is also possibly worth noting that the storage time distribution as defined here cannot be measured using observations, unless suspended particles in transport could be "tracked" and dated in some way.  It is more elegant to determine the ages of particles as they leave a storage reservoir by dating eroding bank deposits, for example. This definition of storage time can actually be defined by field measurements.

Yes, channel length is an important variable to consider. While we did not explicitly highlight it in Figure 1a, we did evaluate it in the main text. As this figure is highly schematized, it is

difficult to depict the size of the "real" model domain. However, as stated in Limaye and Lamb (2013), "The extent of the model domain parallel to the valley axis scales with the average meander wavelength and is long enough that the channel curvature integration never spans the entire channel centerline". To better highlight the importance of channel length, we included a schematic of $x_{tran}$ in Figure 1a and, on line 153, state, *"The model domain extends for 125 channel widths in the mean downstream direction, and is unconstrained in the mean cross-stream direction"*.

As we've defined it, storage durations can be measured using time-series of satellite images. Over longer timescales, it should be possible to use fallout radionuclides (e.g., Black et al. 2010) provided that the data can be corrected for inheritance. We've attempted to make this clearer by changing in language on line 147. It now states, *"In order to capture the full range of relevant time and space scales, which are unavailable in existing field observations of lateral migration (Black et al. 2010, Constantine et al. 2014), we used an established numerical model of river meandering (Howard and Knutson 1984, Limaye and Lamb 2013) to derive a process-based probability distribution of storage durations."*

14. Figure 2, panel 2. The range of x_tran quoted by Pizzuto et al. 2017 is much larger than the data illustrated here. This should be noted in the manuscript.

We appreciate this point, though the differences aren't all that notable. Our definition of transport length scales is distinct from the approach used in Pizzuto et al. 2017, but leads to pretty comparable numbers. For example, the mean $x_{tran}$ we use is 100 km, which means that sediments are deposited and eroded on average 10 times as they transit down a 1000 km long channel. This compares well with what Pizzuto et al. estimate for 10 storage/erosion events for 1000 km of transport distance. However, like Pizzuto et al. (2017), we expect different river systems to have different transport length scales and we use a mean value only for illustrative purposes. As indicated in a previous response, we added a statement about variable transport length scales in Pizzuto et al. (2017) on line 266.

15. Figure 3. It is odd to show the storage duration data in red, but then present the legend associated with these data in black. Please keep the color scheme consistent.

Helpful point!  This color-coding issue have been addressed.

16. Figure 6. Great figure!

We thank the reviewer for this support and note that reviewer #1 had the opposite opinion of this figure. Based on both reviewer comments as well as the opinions of all co-authors, we have opted to keep Figure 6 and modify the associated section 4.2 to address reviewer #1's concerns.